# MULTIMODAL CLASSIFICATION VIA TOTAL CORRELATION MAXIMIZATION

**Feng Yu**[*] **Xiangyu Wu**[*] **Yang Yang**[†] **Jianfeng Lu**[†]
Nanjing University of Science and Technology

## ABSTRACT

Multimodal learning integrates data from diverse sensors to effectively harness information from different modalities. However, recent studies reveal that joint learning often overfits certain modalities while neglecting others, leading to performance inferior to that of unimodal learning. Although previous efforts have sought to balance modal contributions or combine joint and unimodal learning—thereby mitigating the degradation of weaker modalities with promising outcomes—few have examined the relationship between joint and unimodal learning from an information-theoretic perspective. In this paper, we theoretically analyze modality competition and propose a method for multimodal classification by maximizing the total correlation between multimodal features and labels. By maximizing this objective, our approach alleviates modality competition while capturing inter-modal interactions via feature alignment. Building on Mutual Information Neural Estimation (MINE), we introduce **T**otal **C**orrelation **N**eural **E**stimation (**TCNE**) to derive a lower bound for total correlation. Subsequently, we present TCMax, a hyperparameter-free loss function that maximizes total correlation through variational bound optimization. Extensive experiments demonstrate that TCMax outperforms state-of-the-art joint and unimodal learning approaches. Our code is available at https://github.com/hubaak/TCMax.

## 1 INTRODUCTION

Humans better perceive the world through diverse sensory inputs, *i.e.*, text, audio, and vision. Likewise, multimodal fusion models Yang et al. (2021); Yao & Mihalcea (2022); Li et al. (2023b); Zhang et al. (2024); Wei et al. (2024); Zong et al. (2024), which integrate different modalities, are expected to learn more robust and generalized representations than unimodal counterparts. However, recent studies Wang et al. (2020); Huang et al. (2022); Peng et al. (2022) uncover an intriguing phenomenon in multimodal classification: the best-performing unimodal network surpasses the joint learning network, which can be attributed to the differences in convergence and generalization rates among modalities. In such inconsistent convergence states cases, some dominant modalities are adequately overfitted to the training data, causing the multimodal model to overly rely on dominant modalities while neglecting others, ultimately resulting in suboptimal performance.

To address it, several studies Peng et al. (2022); Xu et al. (2023); Li et al. (2023a); Fan et al. (2023); Wei et al. (2024) are committed to balancing multimodal joint learning. Representatively, OGM-GE Peng et al. (2022) modulates the gradient of modality-specific encoders according to their contribution to prediction, inhibiting modalities that converge faster. AGM Li et al. (2023a) dynamically adjusts the gradient contributions from different modalities. However, modality competition Huang et al. (2022) points out that despite joint learning allowing for modality interaction, it easily causes the model to saturate dominant modalities prematurely, neglecting unimodal features that are difficult to learn but conducive to generalization. Followed by some works are proposed to harness the benefits of the unimodal learning strategy, e.g., QMF Zhang et al. (2023) explicitly incorporates a unimodal loss and a regularization term, evaluating the quality of the truncated samples into the loss function. MLA Zhang et al. (2024) decomposes joint learning into alternating unimodal learning, with a lightweight shared head for modality interaction. MMPareto Wei & Hu (2024)

---

[*]Equal contribution.
[†]Corresponding author.

considers both the direction and magnitude of gradients, ensuring that unimodal gradients do not interfere with multimodal training.

Despite significant progress in existing methods, current approaches still primarily rely on joint learning or a combination of joint learning and unimodal learning, with little consideration given to the inherent alignment properties of multimodal data. However, directly integrating joint learning loss, unimodal loss, and alignment loss (i.e., contrastive learning loss) with inherently conflicting optimization objectives requires introducing extra hyperparameters, additional structures, and algorithmic procedures to balance their contributions during training. In this paper, through an information-theoretic analysis, we demonstrate that maximizing the total correlation between the features encoded by each modal encoder in the multimodal model and the labels avoids modality competition while learning inter-modal interactions and incorporating alignment between modalities.

Inspired by Mutual Information Neural Estimation Belghazi et al. (2018), we propose the TCMax loss, which employs Total Correlation Neural Estimation to estimate the lower bound of total correlation. By maximizing this lower bound, we enhance the total correlation between features and labels. We theoretically demonstrate that the output of the model optimized by the TCMax loss possesses the same mathematical significance as that of a model trained using joint learning. Consequently, our method does not require the introduction of additional hyperparameter or structural modifications. Merely employing the TCMax loss during the training phase suffices to achieve favorable results. In summary, our contributions are as follows:

- From an information-theoretic perspective, we elucidate the underlying causes of modality competition and propose that maximizing the total correlation between multimodal features and labels can amalgamate the advantages of joint learning and unimodal learning while incorporating inter-modal alignment
- We introduce Total Correlation Neural Estimation and, based on this, propose the TCMax loss. Theoretically, we prove that optimizing the TCMax loss can increase total correlation and demonstrate that models utilizing TCMax are capable of estimating the joint distribution of multimodal data and the label.
- Comprehensive experiments showcase the considerable improvement over previous joint and unimodal learning methods on various multimodal datasets.

## 2 RELATED WORKS

**Modality Imbalance** Integrating information from multimodal data is essential for comprehensively addressing and solving real-world problems Yu et al. (2025); Wu et al. (2025a;b). However, training a multimodal model using a joint learning strategy is challenging because different modalities often exhibit varying data distributions, require distinct network architectures, and have different convergence rates Wang et al. (2020). Simultaneously, joint learning makes all modalities contribute to one learning objective, causing weak modalities to be suppressed after strong modalities converge, resulting in modality competition Huang et al. (2022). Several works Peng et al. (2022); Zong et al. (2024); Wei et al. (2024); Xu et al. (2023); Li et al. (2023a) have recently been suggested to balance modalities. Representatively, OGM-GE Peng et al. (2022) and AGM Li et al. (2023a) propose balanced multimodal learning methods that correct the contribution imbalance of different modalities by encouraging intensive gradient updating from suboptimal modalities. However, rebalance methods are not able to overcome modality laziness Du et al. (2023) and fail to exploit unimodal features efficiently. Some works Zhang et al. (2023); Wei et al. (2024); Zhang et al. (2024); Wei & Hu (2024) explicitly or implicitly incorporate unimodal loss into their loss functions to avoid modality laziness. Specifically, MLA Zhang et al. (2024) decomposes the conventional multimodal joint optimization scenario into an alternating unimodal learning scenario and exchanges information using a shared head for different modalities. ReconBoost Hua et al. (2024) alternates between different modalities during the learning process, mitigating the issue of synchronous optimization limitations. MMPareto Wei & Hu (2024) balances the objectives of joint learning and unimodal learning using the Pareto method. By preventing modality laziness in multimodal learning, they achieve performance slightly higher than that of unimodal learning.

**Information Theory with Multimodal Learning** In information theory, mutual information quantifies the correlation between two variables in terms of their distribution. Recent works Belghazi

et al. (2018); Hu et al. (2024) have leveraged neural networks to estimate mutual information, bridging the gap between deep learning and information theory. In contrastive learning, the InfoNCE Oord et al. (2018) loss serves as a lower bound estimate of mutual information, providing a solid theoretical foundation for its use. As more applications Chen et al. (2020b); Radford et al. (2021) demonstrate the effectiveness of the InfoNCE loss, the validity of the underlying mutual information theory has been further validated. For more than two variables, total correlation Watanabe (1960) extends mutual information and serves as a measure of the interdependence among multiple variables. In Hwang et al. (2021), the authors applied total correlation in the Multi-View Representation Learning problem and achieved promising results, demonstrating the utility of total correlation in multi-variable scenarios.

## 3 METHOD

### 3.1 MOTIVATION AND PRELIMINARY

**Problem Formulation.** Consider a multimodal data distribution $(x^{(1)}, \ldots, x^{(M)}, y) \sim \mathbb{P}_{\mathcal{X}^{(1)}, \ldots, \mathcal{X}^{(M)}, \mathcal{Y}}$, where $\mathbb{P}_{\mathcal{X}^{(1)}, \ldots, \mathcal{X}^{(M)}, \mathcal{Y}}$ denotes the joint probability distribution over the modalities and labels of the train set, $\mathcal{X}^{(m)}$ represents the sample space of the $m$-th modality, $\mathcal{Y}$ corresponds to the label space. For each modality $m$, a modality-specific encoder $\psi_{\Theta_m}^{(m)} : \mathcal{X}^{(m)} \to \mathcal{Z}^{(m)}$ maps the input space $\mathcal{X}^{(m)}$ to its corresponding embedding space $\mathcal{Z}^{(m)} = \left\{ \psi_{\Theta_m}^{(m)}(x^{(m)}) \mid x^{(m)} \in \mathcal{X}^{(m)} \right\}$. The embeddings from all modalities are subsequently fed into a prediction head $f_\theta : \mathcal{Z}^{(1)} \times \cdots \times \mathcal{Z}^{(M)} \to \mathbb{R}^{|\mathcal{Y}|}$ to integrate information across different modalities and predict the probability distribution of labels $\hat{p}(\hat{y}|Z) = \text{Softmax}(f_\theta(z^{(1)}, \ldots, z^{(M)}))_{\hat{y}}$, where $\hat{y}$ denotes the predicted label and $\theta$ represents the parameters of the prediction head.

**Joint Learning.** The objective of multimodal joint learning is to minimize the distance cross-entropy between the predicted distribution and the ground truth distribution:

$$\mathcal{L}_{\text{joint}} = \mathbb{E}_{(x^{(1)}, \ldots, x^{(M)}, y) \sim \mathbb{P}_{\mathcal{D}}} [\ell(y, \hat{y})] = \mathbb{E}_{(x^{(1)}, \ldots, x^{(M)}, y) \sim \mathbb{P}_{\mathcal{D}}} [-\log \hat{p}(y|Z)], \quad (1)$$

where $\ell(\cdot, \cdot)$ is the cross-entropy loss and $Z = (z^{(1)}, \ldots, z^{(M)})$ is the multimodal feature. As the distribution of $\hat{y}$ is calculated by $Z$, $l(y, \hat{y})$ in Equation 1 can be seen as the conditional cross-entropy under $Z$, denoted as $l(y, \hat{p}|Z)$. Boudiaf et al. (2020) shows minimizing the conditional cross-entropy $l(y, \hat{p}|Z)$ is equivalent to maximizing the mutual information $I(y; Z)$. This implies, with joint learning strategies Peng et al. (2022); Li et al. (2023a), the multimodal model is trained by maximizing the mutual information between the multimodal feature $Z$ and the label $y$.

To explain the cause of modality imbalance from an information-theoretic perspective, we analyze the scenario involving two modalities (audio and visual) without loss of generality, where $Z = (z^{(a)}, z^{(v)})$. Specifically, the multimodal model trained via joint learning aims to maximize the mutual information:

$$I(y; Z) = I(y; z^{(a)}, z^{(v)}) = I(y; z^{(a)}) + I(y; z^{(v)}|z^{(a)}). \quad (2)$$

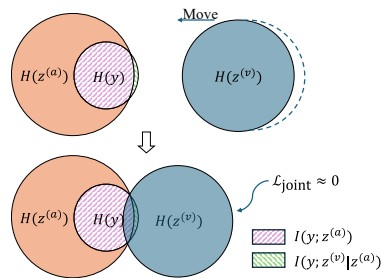

The mutual information between any two variables is bounded by the entropy of either variable. Moreover, since $I(y; z^{(v)}|z^{(a)}) \geq 0$, it follows $H(y) \geq I(y; z^{(a)}, z^{(v)}) \geq I(y; z^{(a)})$. When the encoder of one modality learns faster than the other, assuming $z^{(a)}$ contains sufficient information to predict the label accurately on the train set, then

Figure 1: Venn graph of an extreme case where the audio encoder has already been well-fitted. The visual component (blue) only needs to cover $I(y; z^{(v)]}|z^{(a)})$ to achieve the training objective ($\mathcal{L}_{joint} \approx 0$), therefore ends up being unfitted.

$I(y; z^{(a)})$ in Equation 2 becomes close to $H(y)$. As $I(y; z^{(v)}|z^{(a)}) = I(y; z^{(a)}, z^{(v)}) - I(y; z^{(a)}) \leq H(y) - I(y; z^{(a)})$ and $I(y; z^{(a)}) \approx H(y)$, the upper bound of $I(y; z^{(v)}|z^{(a)})$ tends to be quite small, making it challenging for the visual encoder to learn adequate features though maximizing $I(y; z^{(v)}|z^{(a)})$ as it shows in Figure 1. This phenomenon of resource contention between modalities is termed modality competition Huang et al. (2022).

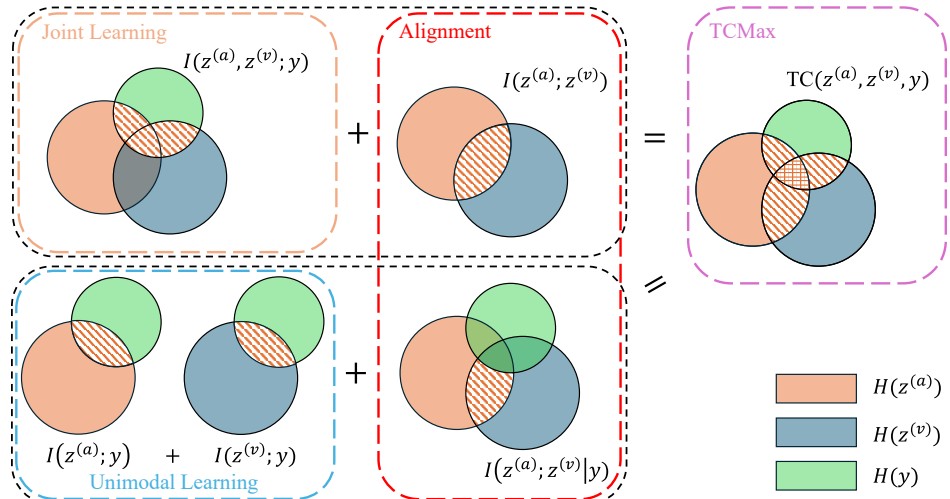

Figure 2: An illustration of the relationship between joint learning, unimodal learning, and learning through maximizing the total correlation.

**Unimodal Learning.** In unimodal learning, each modality-specific model is trained independently and later combined into an ensemble model. Specifically, the logits of the unimodal ensemble model during prediction are equal to the sum of all modality-specific models. For mathematical consistency, we treat the unimodal ensemble model as a single entity during training. However, its prediction head can be decomposed into the sum of modality-specific prediction heads, i.e., $f_\theta(Z) = \sum_m f_{\theta_m}^{(m)}(z^{(m)})$. The optimization objective of unimodal learning is:

$$\mathcal{L}_{\text{unimodal}} = \mathbb{E}_{\left(x^{(1)},\ldots,x^{(M)},y\right)\sim\mathbb{P}_\mathcal{D}} \left[ -\sum_{m=1}^{M} \log \hat{p}^{(m)}(y|z^{(m)}) \right], \tag{3}$$

where $\hat{p}^{(m)}(y|z^{(m)}) = \text{Softmax}(f_\theta^{(m)}(z^{(m)}))_y$ is the predicted distribution of the label of the unimodal model of the $m$-th modality. During training, the unimodal ensemble model maximizes the mutual information $I(y; z^{(m)})$ separately for each modality $m$. For multimodal learning with two modalities, it maximizes:

$$I(y; z^{(a)}) + I(y; z^{(v)}). \tag{4}$$

As mentioned before, we assume the audio modality encoder converges faster, i.e., $z^{(a)}$ captures sufficient information earlier than $z^{(v)}$. Since $I(y; z^{(v)})$ is independent of $z^{(a)}$, it ensures that during the learning process of the visual modality, sufficient mutual information between features and labels can be learned. Although unimodal learning avoids modality competition, its independent training paradigm prevents the model from capturing cross-modal interactions.

While joint learning and unimodal learning focus on modality-label relationships $(\mathcal{X}^{(m)} \leftrightarrow \mathcal{Y})$, multimodal datasets additionally encode cross-modal relationships $(\mathcal{X}^{(i)} \leftrightarrow \mathcal{X}^{(j)})$. To this end, we aim to fully utilize the prior information embedded in $\mathbb{P}_{\mathcal{X}^{(1)},\ldots,\mathcal{X}^{(M)},\mathcal{Y}}$. While mutual information $I(\xi_1; \xi_2)$ effectively measures pairwise dependencies, its multivariate extension $I(\xi_1; \ldots; \xi_n)$ has a limitation: it yields negative values for synergistic interactions. In contrast, total correlation (TC) Watanabe (1960) is non-negative by definition, making it more suitable for measuring multivariate dependencies. Formally:

$$\begin{aligned} \text{TC}(\xi^{(1)}, \xi^{(2)}, \ldots, \xi^{(M)}) &\equiv D_{\text{KL}}\left(\mathbb{P}_{\Xi^{(1)},\ldots,\Xi^{(M)}} \| \mathbb{P}_{\Xi^{(1)}} \times \cdots \times \mathbb{P}_{\Xi^{(M)}}\right) \\ &= \left(\sum_{i=1}^{M} H(\xi^{(i)})\right) - H(\xi^{(1)}, \xi^{(2)}, \ldots, \xi^{(M)}), \end{aligned} \tag{5}$$

where $H$ is the entropy. To leverage the strengths of both joint and unimodal learning, we propose maximizing the TC across all modalities and the label. As Figure 2 shows, in the case of two

modalities, TC can be decomposed as:

$$
\mathrm{TC}(z^{(a)}, z^{(v)}, y) = \begin{cases} \underbrace{I(y; z^{(a)}, z^{(v)})}_{\text{Joint learning}} + \underbrace{I(z^{(a)}; z^{(v)})}_{\text{Alignment}} \\ \underbrace{I(y; z^{(a)}) + I(y; z^{(v)})}_{\text{Unimodal learning}} + \underbrace{I(z^{(a)}; z^{(v)}|y)}_{\text{Alignment}} \end{cases}. \tag{6}
$$

This decomposition reveals that TC simultaneously captures: (1) joint modality-label dependencies (joint learning), (2) modality-modality alignment, and (3) unimodal label dependencies (unimodal learning). This ensures that the model leverages more prior information during training, thereby making the model more robust. In the following sections, we first propose a lower bound estimator for TC, then indirectly maximize TC by optimizing the TCMax loss, which is based on the estimator.

### 3.2 TOTAL CORRELATION NEURAL ESTIMATION

To maximize TC, we propose maximizing its lower bound. This requires a reliable estimator for the TC lower bound. We start from the lower-bound estimator for mutual information. Mutual Information Neural Estimation (MINE) Belghazi et al. (2018) provides a viable approach to estimate a lower bound of mutual information.

**Theorem 1 (MINE Belghazi et al. (2018))** *The mutual information between $Z \in \mathcal{Z}$ and $y \in \mathcal{Y}$ admits the following dual representation:*

$$
I(Z; y) = \sup_{T: \mathcal{Z} \times \mathcal{Y} \to \mathbb{R}} \mathbb{E}_{\mathbb{P}_{\mathcal{Z}, \mathcal{Y}}}[T] - \log\left(\mathbb{E}_{\mathbb{P}_{\mathcal{Z}} \times \mathbb{P}_{\mathcal{Y}}}[e^T]\right), \tag{7}
$$

*where the supremum is taken over all functions $T$ such that the two expectations are finite. As neural networks $T_\theta$ with parameter $\theta \in \Theta$ compose a family of functions which is a subset of $\mathcal{Z} \times \mathcal{Y} \to \mathbb{R}$, we have:*

$$
I(Z; y) \geq \sup_{\theta \in \Theta} \mathbb{E}_{\mathbb{P}_{\mathcal{Z}, \mathcal{Y}}}[T_\theta] - \log\left(\mathbb{E}_{\mathbb{P}_{\mathcal{Z}} \times \mathbb{P}_{\mathcal{Y}}}[e^{T_\theta}]\right). \tag{8}
$$

Fortunately, MINE can be directly extended to Total Correlation Neural Estimation (TCNE). Note that for two variables, TC reduces to mutual information, making MINE a special case of TCNE.

**Corollary 1 (TCNE)** *The total correlation among $M + 1$ variables $z^{(1)} \in \mathcal{Z}^{(1)}, \ldots, z^{(M)} \in \mathcal{Z}^{(M)}$ and $y \in \mathcal{Y}$, admits the following dual representation:*

$$
\mathrm{TC}(z^{(1)}, \ldots, z^{(M)}, y) = \sup_{T: \Omega \to \mathbb{R}} \mathbb{E}_{\mathbb{P}_{\mathcal{Z}^{(1)}, \ldots, \mathcal{Z}^{(M)}, \mathcal{Y}}}[T] - \log\left(\mathbb{E}_{\mathbb{P}_{\mathcal{Z}^{(1)}} \times \cdots \times \mathbb{P}_{\mathcal{Z}^{(M)}} \times \mathbb{P}_{\mathcal{Y}}}[e^T]\right), \tag{9}
$$

*where the supremum is taken over all functions $T$ such that the two expectations are finite and $\Omega = \mathcal{Z}_1 \times \cdots \times \mathcal{Z}_M \times \mathcal{Y}$. As neural networks $T_\theta$ with parameter $\theta \in \Theta$ compose a family of functions which is a subset of $\Omega \to \mathbb{R}$, we have:*

$$
\mathrm{TC}(z^{(1)}, \ldots, z^{(M)}, y) \geq \sup_{\theta \in \Theta} \mathbb{E}_{\mathbb{P}_{\mathcal{Z}^{(1)}, \ldots, \mathcal{Z}^{(M)}, \mathcal{Y}}}[T_\theta] - \log\left(\mathbb{E}_{\mathbb{P}_{\mathcal{Z}^{(1)}} \times \cdots \times \mathbb{P}_{\mathcal{Z}^{(M)}} \times \mathbb{P}_{\mathcal{Y}}}[e^{T_\theta}]\right). \tag{10}
$$

*See the supplement material for all proofs of corollaries and propositions.*

### 3.3 TCMAX LOSS

To align with the form in Corollary 1, we set $T_\theta(z^{(1)}, \ldots, z^{(M)}, y) = f_\theta(z^{(1)}, \ldots, z^{(M)})_y$, decomposing $f_\theta$ into $|\mathcal{Y}|$ functions of the form $Z^{(1)} \times \cdots \times Z^{(M)} \to \mathbb{R}$. Based on Equation 10, we propose the TCMax loss:

$$
\begin{aligned}
\mathcal{L}_{\text{TCMax}} &= -\mathbb{E}_{\mathbb{P}_{\mathcal{Z}^{(1)}, \ldots, \mathcal{Z}^{(M)}, \mathcal{Y}}}[f_\theta] + \log\left(\mathbb{E}_{\mathbb{P}_{\mathcal{Z}^{(1)}} \times \cdots \times \mathbb{P}_{\mathcal{Z}^{(M)}} \times \mathbb{P}_{\mathcal{Y}}}[e^{f_\theta}]\right) \\
&= -\mathbb{E}_{\mathbb{P}_{\mathcal{X}^{(1)}, \ldots, \mathcal{X}^{(M)}, \mathcal{Y}}}[F_\Theta] + \log\left(\mathbb{E}_{\mathbb{P}_{\mathcal{X}^{(1)}} \times \cdots \times \mathbb{P}_{\mathcal{X}^{(M)}} \times \mathbb{P}_{\mathcal{Y}}}[e^{F_\Theta}]\right),
\end{aligned} \tag{11}
$$

where $F_\Theta(x^{(1)}, \ldots, x^{(M)}, y) = f_\theta\left(\psi_{\Theta_1}^{(1)}(x^{(1)}), \ldots, \psi_{\Theta_M}^{(M)}(x^{(M)})\right)_y$ is the multimodal model, $\Theta = \{\Theta_1, \ldots, \Theta_M, \theta\}$ denotes all parameters of the multimodal model. From the above derivation, we

consider the prediction head as a TC estimator on $\mathcal{Z}_1 \times \cdots \times \mathcal{Z}_M \times \mathcal{Y}$. Similarly, the multimodal model can be regarded as an estimator on $\mathcal{X}_1 \times \cdots \times \mathcal{X}_M \times \mathcal{Y}$. Combine with Equation 10, we have following proposition:

**Proposition 1** *The TC between the input data and labels, the TC between features and labels, and our proposed TCMax loss satisfy the following inequality:*

$$\mathrm{TC}(x^{(1)}, \ldots, x^{(M)}, y) \geq \mathrm{TC}(z^{(1)}, \ldots, z^{(M)}, y) \geq -\mathcal{L}_{\mathrm{TCMax}}. \tag{12}$$

Since $\mathcal{L}_{\mathrm{TCMax}} \geq -\mathrm{TC}(z^{(1)}, \ldots, z^{(M)}, y)$, minimizing $\mathcal{L}_{\mathrm{TCMax}}$ pushes $-\mathcal{L}_{\mathrm{TCMax}}$ upward, thereby increasing the lower bound of $\mathrm{TC}(z^{(1)}, \ldots, z^{(M)}, y)$. Since the distribution of the dataset is determined, $\mathrm{TC}(x^{(1)}, \ldots, x^{(M)}, y)$ is a fixed value and does not vary with model parameters. So far, we have not addressed the mathematical interpretation of the TCMax-trained model's output. Next, we prove that the output of this model possesses the same capability to predict the label distribution as a multimodal model trained with joint learning.

**Proposition 2** *The supremum in in Equation 9 reaches its upper bound if and only if $\mathbb{P}_{\mathcal{Z}^{(1)}, \ldots, \mathcal{Z}^{(M)}, \mathcal{Y}} = \mathbb{G}$, where $\mathbb{G}$ is the Gibbs distribution defined as $\mathrm{d}\mathbb{G} = \frac{e^T}{\mathbb{E}_{\mathbb{Q}}[e^T]} \mathrm{d}\mathbb{Q}$ and $\mathbb{Q} = \mathbb{P}_{\mathcal{Z}^{(1)}} \times \cdots \times \mathbb{P}_{\mathcal{Z}^{(M)}} \times \mathbb{P}_{\mathcal{Y}}$.*

Proposition 2 indicates that when the TC estimator in TCNE is accurate, the estimator can also accurately estimate the joint probability distribution of all variables.

**Proposition 3** *The two inequalities in Equation 12 simultaneously hold as equalities if and only if $\mathbb{P}_{\mathcal{X}^{(1)}, \ldots, \mathcal{X}^{(M)}, \mathcal{Y}} = \hat{\mathbb{G}}$, where $\hat{\mathbb{G}}$ is the Gibbs distribution defined as $\mathrm{d}\hat{\mathbb{G}} = \frac{e^{F_\Theta}}{\mathbb{E}_{\mathbb{Q}}[e^{F_\Theta}]} \mathrm{d}\mathbb{Q}$ and $\mathbb{Q} = \mathbb{P}_{\mathcal{X}^{(1)}} \times \cdots \times \mathbb{P}_{\mathcal{X}^{(M)}} \times \mathbb{P}_{\mathcal{Y}}$.*

**No Modifications when Predicting.** Proposition 3 demonstrates that the lower bound of $\mathcal{L}_{\mathrm{TCMax}}$ is $-\mathrm{TC}(x^{(1)}, \ldots, x^{(M)}, y)$, and when this lower bound is achieved, we have

$$p(y|x^{(1)}, \ldots, x^{(M)}) = \frac{p(y, x^{(1)}, \ldots, x^{(M)})}{\sum_{k \in \mathcal{Y}} p(k, x^{(1)}, \ldots, x^{(M)})} = \frac{e^{F_\Theta(x^{(1)}, \ldots, x^{(M)}, y)}}{\sum_{k \in \mathcal{Y}} e^{F_\Theta(x^{(1)}, \ldots, x^{(M)}, k)}} = \hat{p}_y, \tag{13}$$

where $p$ is the probability mass function of data distribution $\mathbb{P}_{\mathcal{D}}$. Equation 13 indicates that the model trained using the TCMax loss does not require additional operations or modifications to the model structure during prediction. The only difference between our proposed method and joint learning in practice is replacing $\mathcal{L}_{\mathrm{joint}}$ with $\mathcal{L}_{\mathrm{TCMax}}$ during training, yet it yields more robust results.

## 3.4 COMPUTATIONAL COST

When training a multimodal model that includes both audio and visual modalities, a direct implementation of $\mathcal{L}_{\mathrm{TCMax}}$ in a mini-batch $\mathcal{B}$ is:

$$\mathcal{L}_{\mathrm{TCMax}} = -\frac{1}{|\mathcal{B}|} \sum_{i \in \mathcal{B}} \log \frac{\exp f_\theta \left( \psi_{\Theta_a}^{(a)}(x_i^{(a)}), \psi_{\Theta_v}^{(v)}(x_i^{(v)}) \right)_{y_i}}{\sum_{(j,k,y') \in \mathcal{B} \times \mathcal{B} \times \mathcal{Y}} \exp f_\theta \left( \psi_{\Theta_a}^{(a)}(x_j^{(a)}), \psi_{\Theta_v}^{(v)}(x_k^{(v)}) \right)_{y'}} - \log |\mathcal{B}|^2 |\mathcal{Y}|, \tag{14}$$

where $x_i^{(m)}$ denotes the $i$-th sample of $m$-th modality. Notice that using $\mathcal{L}_{\mathrm{TCMax}}$ requires forwarding the prediction head $|\mathcal{B}|^M$ times. Although the parameter count of the prediction head is generally much smaller compared to the encoders, for a large number of modalities $M$ and a large batch size, this can still introduce significant additional computational overhead. To mitigate this overhead, the computation can be optimized by sampling only a certain number of negative samples (denominator) in the feature space, i.e., randomly sampling $\mathcal{N} \subset \mathcal{B} \times \mathcal{B}$, where each $(i, j) \in \mathcal{N}$ is sampled uniformly without replacement from $\mathcal{B} \times \mathcal{B}$. $\mathcal{L}_{\mathrm{TCMax}}$ with sampling is:

$$\mathcal{L}_{\mathrm{TCMax}} = -\frac{1}{|\mathcal{B}|} \sum_{i \in \mathcal{B}} \log \frac{\exp f_\theta \left( \psi_{\Theta_a}^{(a)}(x_i^{(a)}), \psi_{\Theta_v}^{(v)}(x_i^{(v)}) \right)_{y_i}}{\sum_{(j,k) \in \mathcal{N}} \sum_{y' \in \mathcal{Y}} \exp f_\theta \left( \psi_{\Theta_a}^{(a)}(x_j^{(a)}), \psi_{\Theta_v}^{(v)}(x_k^{(v)}) \right)_{y'}} - \log |\mathcal{N}| |\mathcal{Y}|. \tag{15}$$

For linear fusion $f_\theta(z^{(a)}, z^{(v)}) = f_{\theta_a}^{(a)}(z^{(a)}) + f_{\theta_v}^{(v)}(z^{(v)})$, the denominator decouples into separate sums over modalities due to the identity $\exp(a + b) = \exp(a)\exp(b)$. $\mathcal{L}_{\text{TCMax}}$ becomes:

$$\mathcal{L}_{\text{TCMax}} = -\frac{1}{|\mathcal{B}|}\sum_{i\in\mathcal{B}}\log\frac{\exp f_{\theta_a}^{(a)}(\psi_{\Theta_a}^{(a)}(x_i^{(a)}))_{y_i}\exp f_{\theta_v}^{(v)}(\psi_{\Theta_v}^{(v)}(x_i^{(v)}))_{y_i}}{\sum_{y'\in\mathcal{Y}}\left(\sum_{j\in\mathcal{B}}\exp f_{\theta_a}^{(a)}(\psi_{\Theta_a}^{(a)}(x_j^{(a)}))_{y'}\right)\left(\sum_{k\in\mathcal{B}}\exp f_{\theta_v}^{(v)}(\psi_{\Theta_v}^{(v)}(x_k^{(v)}))_{y'}\right)}$$
$$-\log|\mathcal{B}|^2|\mathcal{Y}|. \quad (16)$$

In this way, only $|\mathcal{B}|$ forward passes of the prediction head are required, introducing almost no additional computational overhead with $\mathcal{L}_{\text{TCMax}}$.

## 4 EXPERIMENTS

### 4.1 DATASETS

**CREMA-D** Cao et al. (2014) encompasses $7,442$ audio-visual clips from 91 actors expressing six emotions, with emotion labels determined by $2,443$ crowd-sourced raters. **Kinetics-Sounds** (KS) Arandjelovic & Zisserman (2017), a subset of Kinetics Kay et al. (2017) dataset, includes $19,000$ $10s$ videos across 31 human action labels, annotated manually through Mechanical Turk. **AVE** Tian et al. (2018) focuses on localizing audio-visual events in $4,143$ $10s$ videos across 28 labels, sourced from YouTube with frame-level labeling for both audio and visual components. **VGGSound** Chen et al. (2020a) is a large dataset of 309 labels, with $10s$ videos that exhibit clear audio-visual correlations. It contains $152,638$ training videos and $13,294$ testing videos. **UCF101** Soomro (2012) comprises $13,320$ videos from 101 action labels. The clips, ranging from $3s$ to $10s$, are split into a training set with $9,537$ clips and a testing set with $3,783$ clips, utilizing the official train-test split. **MVSA** Niu et al. (2016) (MVSA-Single) is a multimodal sentiment analysis dataset that jointly leverages text and image data for classification.

### 4.2 IMPLEMENTATION DETAILS

**Backbone and Hyperparameter** For all audio-visual datasets, we follow the same setting as in the previous study Peng et al. (2022), selecting ResNet-18 He et al. (2016) as the encoder for both audio and visual modalities, and training it from scratch. For the audio modality, inputs are converted into spectrograms McFee et al. (2015) of size $129\times862$ for the CREMA-D, AVE, and VGGSound datasets, and fbank Davis & Mermelstein (1980) acoustic features for the Kinetics-Sounds dataset. For the visual modality, we extract images from videos at 1 fps and use 3 images as input for the CREMA-D, Kinetics-Sounds, and AVE datasets, and four images as input for the VGGSound dataset. For the UCF101 dataset, we extract frames and optical flow data from the videos at 1 fps, using 3 RGB frames and 10 optical flow frames for each sample. We use ResNet-18 as the backbone for both RGB and optical flow modalities and train ResNet-18 from scratch as their backbone. We utilize SGD Robbins & Monro (1951) with 0.9 momentum and $1e-4$ weight decay as the optimizer for all experiments. We set (learning rate, mini-batch size, epochs) to $(1e-3, 32, 200)$ for CREMA-D and AVE, $(1e-3, 64, 200)$ for Kinetics-Sounds, $(1e-3, 64, 100)$ for VGGSound, and $(1e-3, 32, 400)$ for UCF101. All of our experiments were performed on one NVIDIA Tesla V100 GPU.

### 4.3 COMPARISON WITH STATE-OF-THE-ARTS

**Compared methods** We conduct comprehensive comparisons of TCMax with several baselines and recent studies. (1) Baselines: concatenation (Concat), share predicted head (Share Head), unimodal fusion (Unimodal); (2) Recent studies: FiLM Perez et al. (2018), BiGated Kiela et al. (2018), OGM-GE Peng et al. (2022), OPM Wei et al. (2024), QMF Zhang et al. (2023), AGM Li et al. (2023a), MLA Zhang et al. (2024), MMPareto Wei & Hu (2024).

#### 4.3.1 RESULTS OF TEST ACCURACY

Table 1 presents testing accuracy of using a single modality and combining modalities (*i.e.*, Multi). *First*, joint learning shows severe modality imbalance, with one modality significantly underperforming, and generally yields worse results than alternatives. *Second*, balanced joint learning methods

Table 1: Results of the average test accuracy(%) of three random seeds on CREMA-D, Kinetics-Sounds, AVE, VGGSound, and UCF101 datasets. Both the results of only using a single modality ("Audio" and "Visual") and the results of combining all modalities ("Multi") are listed. The best results and second best results are **bold** and underlined, respectively.

| Methods | CREMA-D | | | Kinetics-Sounds | | | AVE | | | VGGSound | | | UCF101 | | |
|---|---|---|---|---|---|---|---|---|---|---|---|---|---|---|---|
| | Audio | Visual | **Multi** | Audio | Visual | **Multi** | Audio | Visual | **Multi** | Audio | Visual | **Multi** | RGB | OF | **Multi** |
| Concat | 58.6 | 26.5 | 68.5 | 36.8 | 22.2 | 53.5 | 48.9 | 17.2 | 60.4 | 29.5 | 10.3 | 41.1 | 27.5 | 15.8 | 46.1 |
| Share Head | 63.0 | 29.5 | 69.3 | 38.7 | 29.1 | 53.7 | 49.7 | 22.4 | 61.4 | 30.9 | 13.0 | 41.8 | 31.2 | 20.2 | 46.0 |
| FiLM Perez et al. (2018) | - | - | 65.3 | - | - | 52.6 | - | - | 58.8 | - | - | 38.3 | - | - | 44.7 |
| BiGated Kiela et al. (2018) | - | - | 64.7 | - | - | 49.1 | - | - | 59.3 | - | - | 38.3 | - | - | 46.7 |
| OGM-GE Peng et al. (2022) | 55.8 | 45.8 | 75.2 | 35.7 | 25.6 | 55.4 | 39.5 | 19.3 | 61.3 | 28.7 | 13.2 | 42.5 | 23.3 | 16.6 | 44.2 |
| AGM Li et al. (2023a) | 61.4 | 39.0 | 71.2 | 36.4 | 29.4 | 57.7 | 44.2 | 19.3 | 61.4 | 30.5 | 14.7 | 44.7 | 27.2 | 17.6 | 46.0 |
| Unimodal Ensemble | 62.9 | 74.9 | 82.1 | 43.0 | 45.8 | 62.3 | 54.1 | 36.7 | 62.9 | 34.8 | 26.7 | 46.3 | 39.2 | 29.2 | 51.6 |
| QMF Zhang et al. (2023) | **65.5** | 74.3 | 81.4 | 44.6 | 44.2 | 62.1 | **55.1** | 37.0 | 65.1 | 34.6 | 26.1 | 45.8 | 39.2 | 31.2 | 52.1 |
| OPM Wei et al. (2024) | 61.3 | 67.2 | 79.4 | 39.1 | 44.1 | 61.7 | 48.8 | 36.2 | 63.1 | 31.2 | 23.5 | 45.9 | 38.6 | 26.6 | 50.8 |
| OGM-GE + OPM Wei et al. (2024) | 58.3 | 69.8 | 80.3 | 37.6 | 44.1 | 62.5 | 48.2 | 35.3 | 63.3 | 31.6 | 23.5 | 45.5 | 38.6 | 26.9 | 48.3 |
| MLA Zhang et al. (2024) | 63.5 | 75.6 | 81.0 | 41.6 | 44.9 | 61.1 | 53.2 | 37.9 | 62.6 | 35.3 | 25.8 | 44.5 | 40.1 | 31.0 | 51.2 |
| MMPareto Wei & Hu (2024) | 65.4 | **75.7** | 74.4 | **44.8** | **49.4** | 62.7 | 54.0 | **41.1** | 63.1 | 35.5 | 27.5 | 46.2 | **42.4** | 33.0 | **55.9** |
| **Ours**Concat | 63.7 | 75.0 | **82.8** | 41.0 | 41.4 | 62.4 | 53.8 | 33.5 | 63.2 | 33.9 | 22.0 | **47.6** | 37.7 | 32.5 | 55.4 |
| **Ours**Share Head | 63.0 | 73.6 | 82.7 | 43.4 | 43.3 | **63.5** | 54.2 | 36.7 | 64.5 | 36.3 | 24.3 | 47.5 | 41.0 | **37.2** | 56.0 |

Table 2: Results of Jensen–Shannon divergence between predictions of two modalities on CREMA-D, Kinetics-Sounds, AVE, VGGSound, and UCF101 datasets. The minima and second minima results are **bold** and underlined, respectively.

| Dataset | Concat | Share Head | OGM-GE | AGM | Unimodal | QMF | OPM | MLA | MMPareto | **Ours**Concat | **Ours**Share Head |
|---|---|---|---|---|---|---|---|---|---|---|---|
| CREMA-D | 0.478 | 0.490 | 0.518 | 0.400 | 0.312 | 0.293 | 0.337 | 0.306 | 0.314 | 0.284 | **0.271** |
| Kinetics-Sounds | 0.543 | 0.560 | 0.551 | 0.499 | 0.459 | 0.455 | 0.448 | 0.466 | 0.440 | 0.423 | **0.390** |
| AVE | 0.562 | 0.568 | 0.560 | 0.540 | 0.462 | 0.471 | 0.470 | 0.468 | 0.465 | 0.452 | **0.406** |
| VGGSound | 0.620 | 0.631 | 0.610 | 0.594 | 0.526 | 0.528 | 0.513 | 0.528 | 0.539 | 0.513 | **0.473** |
| UCF101 | 0.584 | 0.579 | 0.589 | 0.586 | 0.463 | 0.531 | 0.473 | 0.457 | 0.485 | 0.427 | **0.366** |

(e.g., OGM-GE, AGM) improve the weaker modality but slightly degrade the stronger one, failing to consistently surpass unimodal-loss methods. *Third*, unimodal-based methods prevent overfitting and sometimes outperform pure unimodal learning, while achieving the highest single-modal accuracy. *Finally*, TCMax achieves the best multimodal scores, but its single-modality performance matches other unimodal-based methods—suggesting its gains stem from cross-modal synergy rather than individual improvements.

### 4.3.2 RESULTS OF JENSEN–SHANNON DIVERGENCE

To investigate the correlation of prediction outcomes across modalities, we calculate the average Jensen–Shannon divergence (JS-divergence) between the prediction results of two separate modalities in Table 2. This metric quantifies the degree of correlation in predictions between the two modalities, with a lower JS-divergence signifying a stronger correlation. As shown in the table, our TCMax consistently achieves the smallest JS-divergence for the single-modal predictions across all datasets, indicating that TCMax which is rooted in contrastive learning, facilitates the model to learn cross-modal representations, thereby enhancing the correlation of predictions.

### 4.4 FURTHER ANALYSIS

**Number of Sampled Negative Pairs.** We first explore the effect of sampling different numbers of negative pairs, as mentioned in Equation 15. Figure 3a shows that, for the CREMA-D dataset, the accuracy of TCMax rises with an increase in the number of negative pair samplings, achieving optimal performance at 1024. In contrast, on the UCF101 dataset, the best performance is observed at the maximum sampling number 256.

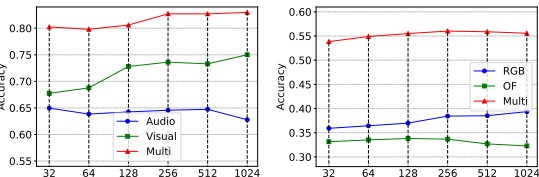

(a) CREMA-D, Batchsize 64    (b) UCF101, Batchsize 32

Figure 3: Accuracy on different numbers of sampled negative pairs.

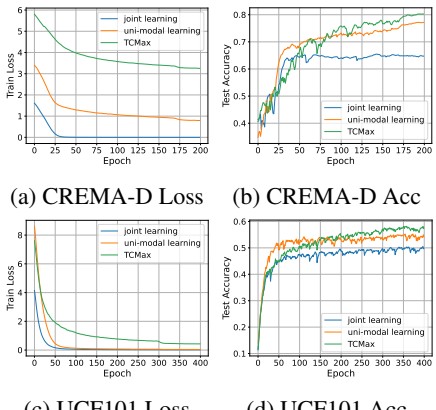

(a) CREMA-D Loss  (b) CREMA-D Acc

(c) UCF101 Loss  (d) UCF101 Acc

Figure 4: Train loss and test accuracy of joint learning, unimodal learning, and TCMax on CREMA-D and UCF101 datasets.

|  |  | Concat | Share Head | Unimodal | $\textbf{Ours}_{Concat}$ |
|---|---|---|---|---|---|
| CREMA-D | $H_{(A)}$ | 0.369 | 0.184 | 0.320 | 0.575 |
|  | $H_{(V)}$ | 1.076 | 1.229 | 0.746 | 0.890 |
|  | $\rho$ | 2.913 | 6.674 | 2.331 | **1.549** |
| UCF101 | $H_{(RGB)}$ | 1.245 | 0.630 | 0.921 | 2.000 |
|  | $H_{(OF)}$ | 2.244 | 1.170 | 1.259 | 2.265 |
|  | $\rho$ | 1.802 | 1.856 | 1.368 | **1.132** |

Table 3: Results of average entropy of predictions by single modality on test sets of CREMA-D and UCF101 datasets. $H_{(M)}$ denotes the entropy of predictions of the 'M' modality, and $\rho$ represents the ratio of the entropy of the weak modality to the entropy of the strong modality. For CREMA-D and UCF101 datasets, $\rho = H_{(V)}/H_{(A)}$ and $\rho = H_{(OF)}/H_{(RGB)}$, respectively.

**TCMax Prevents Overfitting.** Here, we visualize how TCMax effectively mitigates the risk of overfitting. As depicted in Figure 4, on both the CREMA-D and UCF101 datasets, TCMax loss remains consistently higher than that for joint and unimodal learning, which prevents the model parameters from updating at all. Although TCMax exhibits inferior performance compared to unimodal learning in the early stages of training, as the training progresses into the middle stage, TCMax begins to gradually presents its strengths and ultimately converges to a stable performance level.

**Average Entropy of Predictions.** In Table 3, we compute the average entropy predictions by single modality and the ratio between strong and weak modalities. This reflects the model's ability to balance predictions across different modalities. Typically, a lower ratio indicates a more equitable contribution from both modalities. The table reveals that our method successfully achieves a balanced representation of the various modalities.

**Analysis with Pretrained Encoders** As shown in Table 4, we adopt CLIP Radford et al. (2021) as the frozen feature encoder for both image and text modalities on the MVSA dataset. During training, only the multimodal classifier is optimized while keeping the encoder parameters fixed. Our results show that: (1) Joint learning outperforms unimodal learning because the limited parameter space prevents overfitting; (2) TCMax maintains competitive performance by effectively modeling cross-modal interactions, similar to joint learning.

Table 4: Result of the average test accuracy(%) of 10 random seeds on MVSA with frozen CLIP pretrained encoders.

| Methods | RN50 | | | ViT-B/32 | | |
|---|---|---|---|---|---|---|
|  | Image | Text | **Multi** | Image | Text | **Multi** |
| Joint | 75.76 | 73.60 | 81.23 | 76.88 | 74.27 | 82.83 |
| Unimodal | **76.74** | **77.16** | 80.02 | **78.54** | **76.97** | 81.77 |
| TCMax | 75.38 | 74.97 | **81.75** | 78.03 | 76.55 | **84.05** |

## 5 DISCUSSION

**Conclusion** This study investigates the causes of modality imbalance in multimodal classification tasks from an information-theoretic perspective and proposes a learning objective to maximize total correlation to fully utilize cross-modal information in multimodal datasets. We propose the Total Correlation Neural Estimation(TCNE), which employs neural networks to estimate the lower bound of total correlation. Building on this theoretical foundation, we introduce a parameter-free loss function, TCMax, for multimodal classification tasks. By maximizing total correlation, our approach enables more comprehensive utilization of prior information in multimodal datasets, thereby achieving enhanced robustness. Comparative experiments with state-of-the-art methods demonstrate the effectiveness of our approach across multiple multimodal classification benchmarks.

**Limitation** The current TCMax framework is primarily designed for classification tasks and cannot be directly extended to other multimodal applications such as multimodal object detection or generative tasks. Successful adaptation to these domains would require explicit definitions of input-output probability distributions. Furthermore, while TCMax establishes a foundational multimodal learning paradigm, its full potential depends on developing model architectures specifically optimized for this framework.

ACKNOWLEDGEMENTS

This work is supported by the NSFC (62276131, 62506168), Natural Science Foundation of Jiangsu Province of China under Grant (BK20240081, BK20251431).

The authors gratefully acknowledge financial support from the China Scholarship Council (CSC) (Grant No. 202506840036).

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

# A  PROOFS

## A.1  PROOF OF COROLLARY 1

**Corollary 2 (Corollary 1 restated, TCNE)** *The total correlation between $M + 1$ variables $z^{(1)} \in \mathcal{Z}^{(1)}, \ldots, z^{(M)} \in \mathcal{Z}^{(M)}$ and $y \in \mathcal{Y}$, admits the following dual representation:*

$$\mathrm{TC}(z^{(1)}, \ldots, z^{(M)}, y) = \sup_{T:\Omega\to\mathbb{R}} \mathbb{E}_{\mathbb{P}_{\mathcal{Z}^{(1)},\ldots,\mathcal{Z}^{(M)},\mathcal{Y}}}[T] - \log\left(\mathbb{E}_{\mathbb{P}_{\mathcal{Z}^{(1)}}\times\cdots\times\mathbb{P}_{\mathcal{Z}^{(M)}}\times\mathbb{P}_{\mathcal{Y}}}\left[e^T\right]\right) \quad (17)$$

*where the supremum is taken over all functions $T$ such that the two expectations are finite and $\Omega = \mathcal{Z}_1 \times \cdots \times \mathcal{Z}_M \times \mathcal{Y}$. As neural networks $T_\theta$ with parameter $\theta \in \Theta$ composed a family of functions which is a subset of $\Omega \to \mathbb{R}$, we have:*

$$\mathrm{TC}(z^{(1)}, \ldots, z^{(M)}, y) \geq \sup_{\theta\in\Theta} \mathbb{E}_{\mathbb{P}_{\mathcal{Z}^{(1)},\ldots,\mathcal{Z}^{(M)},\mathcal{Y}}}[T_\theta] - \log\left(\mathbb{E}_{\mathbb{P}_{\mathcal{Z}^{(1)}}\times\cdots\times\mathbb{P}_{\mathcal{Z}^{(M)}}\times\mathbb{P}_{\mathcal{Y}}}\left[e^{T_\theta}\right]\right) \quad (18)$$

We follow the proof in the paper of MINE Belghazi et al. (2018) to prove TCNE. First, we begin with the Donsker-Varadhan representation theorem.

**Theorem 2** *(The Donsker-Varadhan representation Donsker & Varadhan (1983)) The KL divergence admits the following dual representation:*

$$D_{KL}(\mathbb{P}\|\mathbb{Q}) = \sup_{T:\Omega\to\mathbb{R}} \mathbb{E}_{\mathbb{P}}[T] - \log\left(\mathbb{E}_{\mathbb{Q}}[e^T]\right), \quad (19)$$

*where the supremum is taken over all functions $T$ such that the two expectations are finite.*

For a given function $T$, consider the Gibbs distribution defined by $d\mathbb{G} = \frac{1}{Z}e^T d\mathbb{Q}$, where $Z = \mathbb{E}_{\mathbb{Q}}[e^T]$ is he partition function and $T$ servers as the energy function in the Gibbs distribution. The right hand of Equation 19 can be written as:

$$\mathbb{E}_{\mathbb{P}}[T] - \log\left(\mathbb{E}_{\mathbb{Q}}[e^T]\right) = \mathbb{E}_{\mathbb{P}}[T] - \log Z = \mathbb{E}_{\mathbb{P}}\left[\log\frac{e^T}{Z}\right] = \mathbb{E}_{\mathbb{P}}\left[\log\frac{d\mathbb{G}}{d\mathbb{Q}}\right]. \quad (20)$$

Let $\Delta$ be the gap:

$$\Delta \equiv D_{KL}(\mathbb{P}\|\mathbb{Q}) - \mathbb{E}_{\mathbb{P}}[T] - \log\left(\mathbb{E}_{\mathbb{Q}}[e^T]\right), \quad (21)$$

with Equation 20, we can write $\Delta$ as KL-divergence:

$$\Delta = \mathbb{E}_{\mathbb{P}}\left[\log\frac{d\mathbb{P}}{d\mathbb{Q}} - \log\frac{d\mathbb{G}}{d\mathbb{Q}}\right] = \mathbb{E}_{\mathbb{P}}\left[\log\frac{d\mathbb{P}}{d\mathbb{G}}\right] = D_{KL}(\mathbb{P}\|\mathbb{G}). \quad (22)$$

The positivity of the KL-divergence gives $\Delta \geq 0$. We have thus shown that for any $T$,

$$D_{KL}(\mathbb{P}\|\mathbb{Q}) \geq \mathbb{E}_{\mathbb{P}}[T] - \log\left(\mathbb{E}_{\mathbb{Q}}[e^T]\right), \quad (23)$$

The inequality is preserved upon taking the supremum over the right-hand side. The bound is tight when $\mathbb{G} = \mathbb{P}$, namely for optimal functions $T^*$ taking over the form $T^* = \log\frac{d\mathbb{P}}{d\mathbb{Q}} + Const$ for some constant $Const \in \mathbb{R}$.

To prove Equation 17 in Corollary 2, we replace $\mathbb{P}$ and $\mathbb{Q}$ with $\mathbb{P}_{\mathcal{Z}^{(1)},\ldots,\mathcal{Z}^{(M)},\mathcal{Y}}$ and $\mathbb{P}_{\mathcal{Z}^{(1)}} \times \cdots \times \mathbb{P}_{\mathcal{Z}^{(M)}} \times \mathbb{P}_{\mathcal{Y}}$ in Equation 19 so we have:

$$D_{KL}\left(\mathbb{P}_{\mathcal{Z}^{(1)},\ldots,\mathcal{Z}^{(M)},\mathcal{Y}}\|\mathbb{P}_{\mathcal{Z}^{(1)}} \times \cdots \times \mathbb{P}_{\mathcal{Z}^{(M)}} \times \mathbb{P}_{\mathcal{Y}}\right) =$$
$$\sup_{T:\Omega\to\mathbb{R}} \mathbb{E}_{\mathbb{P}_{\mathcal{Z}^{(1)},\ldots,\mathcal{Z}^{(M)},\mathcal{Y}}}[T] - \log\left(\mathbb{E}_{\mathbb{P}_{\mathcal{Z}^{(1)}}\times\cdots\times\mathbb{P}_{\mathcal{Z}^{(M)}}\times\mathbb{P}_{\mathcal{Y}}}[e^T]\right). \quad (24)$$

With the KL-divergence form of total correlation:

$$\mathrm{TC}(z^{(1)}, \ldots, z^{(M)}, y) = D_{KL}\left(\mathbb{P}_{\mathcal{Z}^{(1)},\ldots,\mathcal{Z}^{(M)},\mathcal{Y}}\|\mathbb{P}_{\mathcal{Z}^{(1)}} \times \cdots \times \mathbb{P}_{\mathcal{Z}^{(M)}} \times \mathbb{P}_{\mathcal{Y}}\right), \quad (25)$$

we can proof the Equation 17 in Corollary 2. As neural networks $T_\theta$ with parameter $\theta \in \Theta$ belongs to $\{T | T : \Omega \to \mathbb{R}\}$, the supremum taken over all networks $T_\theta$ is less than or equal to the supremum taken over all functions $T$,

$$\mathrm{TC}(z^{(1)}, \ldots, z^{(M)}, y) = \sup_{T:\Omega\to\mathbb{R}} \mathbb{E}_{\mathbb{P}_{\mathcal{Z}^{(1)},\ldots,\mathcal{Z}^{(M)},\mathcal{Y}}}[T] - \log\left(\mathbb{E}_{\mathbb{P}_{\mathcal{Z}^{(1)}}\times\cdots\times\mathbb{P}_{\mathcal{Z}^{(M)}}\times\mathbb{P}_{\mathcal{Y}}}\left[e^T\right]\right)$$
$$\geq \sup_{\theta\in\Theta} \mathbb{E}_{\mathbb{P}_{\mathcal{Z}^{(1)},\ldots,\mathcal{Z}^{(M)},\mathcal{Y}}}[T_\theta] - \log\left(\mathbb{E}_{\mathbb{P}_{\mathcal{Z}^{(1)}}\times\cdots\times\mathbb{P}_{\mathcal{Z}^{(M)}}\times\mathbb{P}_{\mathcal{Y}}}\left[e^{T_\theta}\right]\right). \quad (26)$$

Thus we prove Corollary 2.

## A.2 PROOF OF PROPOSITION 1

**Proposition 4 (Proposition 1 restated)** *The TC between the input data and labels, the TC between features and labels, and our proposed TCMax loss satisfy the following inequality:*

$$\mathrm{TC}(x^{(1)}, \ldots, x^{(M)}, y) \geq \mathrm{TC}(z^{(1)}, \ldots, z^{(M)}, y) \geq -\mathcal{L}_{\mathrm{TCMax}} \tag{27}$$

We first consider the TCNE form of $\mathrm{TC}(x^{(1)}, \ldots, x^{(M)}, y)$ and $\mathrm{TC}(z^{(1)}, \ldots, z^{(M)}, y)$:

$$\mathrm{TC}(x^{(1)}, \ldots, x^{(M)}, y) = \sup_{T_{\mathcal{X}}:\Omega_{\mathcal{X}}\to\mathbb{R}} \mathbb{E}_{\mathbb{P}_{\mathcal{X}^{(1)},\ldots,\mathcal{X}^{(M)},\mathcal{Y}}} \left[T_{\mathcal{X}}\right] - \log\left(\mathbb{E}_{\mathbb{P}_{\mathcal{X}^{(1)}}\times\cdots\times\mathbb{P}_{\mathcal{X}^{(M)}}\times\mathbb{P}_{\mathcal{Y}}} \left[e^{T_{\mathcal{X}}}\right]\right), \tag{28}$$

$$\mathrm{TC}(z^{(1)}, \ldots, z^{(M)}, y) = \sup_{T_{\mathcal{Z}}:\Omega_{\mathcal{Z}}\to\mathbb{R}} \mathbb{E}_{\mathbb{P}_{\mathcal{Z}^{(1)},\ldots,\mathcal{Z}^{(M)},\mathcal{Y}}} \left[T_{\mathcal{Z}}\right] - \log\left(\mathbb{E}_{\mathbb{P}_{\mathcal{Z}^{(1)}}\times\cdots\times\mathbb{P}_{\mathcal{Z}^{(M)}}\times\mathbb{P}_{\mathcal{Y}}} \left[e^{T_{\mathcal{Z}}}\right]\right), \tag{29}$$

where $\Omega_{\mathcal{X}}$ and $\Omega_{\mathcal{Z}}$ are the input space (including the label) and the embedding space, respectively. As modality-specific encoders are used to extract the embedding $z^{(1,\cdots,M)}$ from the input $x^{(1,\cdots,M)}$, a fix function $\Psi : \Omega_{\mathcal{X}} \to \Omega_{\mathcal{Z}}$ is defined here. So we can rewrite Equation 29:

$$\mathrm{TC}(z^{(1)}, \ldots, z^{(M)}, y) = \sup_{T_{\mathcal{Z}}:\Omega_{\mathcal{Z}}\to\mathbb{R}} \mathbb{E}_{\mathbb{P}_{\mathcal{X}^{(1)},\ldots,\mathcal{X}^{(M)},\mathcal{Y}}} \left[\Psi \circ T_{\mathcal{Z}}\right] - \log\left(\mathbb{E}_{\mathbb{P}_{\mathcal{X}^{(1)}}\times\cdots\times\mathbb{P}_{\mathcal{X}^{(M)}}\times\mathbb{P}_{\mathcal{Y}}} \left[e^{\Psi \circ T_{\mathcal{X}}}\right]\right). \tag{30}$$

Since $\{\Psi \circ T_{\mathcal{Z}} | T_{\mathcal{Z}} : \Omega_{\mathcal{Z}} \to \mathbb{R}\}$ is a subset of $\{T_{\mathcal{X}} | T_{\mathcal{X}} : \Omega_{\mathcal{X}} \to \mathbb{R}\}$, the supremum in Equation 30 is not surpass than the supremum in Equation 28. Thus, we prove the first inequality in the proposition.

We first consider the form of TCMax loss to prove he second inequality:

$$\mathcal{L}_{\mathrm{TCMax}} = -\mathbb{E}_{\mathbb{P}_{\mathcal{Z}^{(1)},\ldots,\mathcal{Z}^{(M)},\mathcal{Y}}} \left[f_\theta\right] + \log\left(\mathbb{E}_{\mathbb{P}_{\mathcal{Z}^{(1)}}\times\cdots\times\mathbb{P}_{\mathcal{Z}^{(M)}}\times\mathbb{P}_{\mathcal{Y}}} \left[e^{f_\theta}\right]\right). \tag{31}$$

As the predicted head $f_\theta$ is a special case in $T_{\mathcal{Z}} | T_{\mathcal{Z}} : \Omega_{\mathcal{Z}} \to \mathbb{R}$, therefore:

$$\begin{aligned}
\mathrm{TC}(z^{(1)}, \ldots, z^{(M)}, y) &= \sup_{T_{\mathcal{Z}}:\Omega_{\mathcal{Z}}\to\mathbb{R}} \mathbb{E}_{\mathbb{P}_{\mathcal{Z}^{(1)},\ldots,\mathcal{Z}^{(M)},\mathcal{Y}}} \left[T_{\mathcal{Z}}\right] - \log\left(\mathbb{E}_{\mathbb{P}_{\mathcal{Z}^{(1)}}\times\cdots\times\mathbb{P}_{\mathcal{Z}^{(M)}}\times\mathbb{P}_{\mathcal{Y}}} \left[e^{T_{\mathcal{Z}}}\right]\right) \\
&\geq \mathbb{E}_{\mathbb{P}_{\mathcal{Z}^{(1)},\ldots,\mathcal{Z}^{(M)},\mathcal{Y}}} \left[f_\theta\right] - \log\left(\mathbb{E}_{\mathbb{P}_{\mathcal{Z}^{(1)}}\times\cdots\times\mathbb{P}_{\mathcal{Z}^{(M)}}\times\mathbb{P}_{\mathcal{Y}}} \left[e^{f_\theta}\right]\right) = -\mathcal{L}_{\mathrm{TCMax}},
\end{aligned} \tag{32}$$

thus, we prove the second inequality in the proposition.

## A.3 PROOF OF PROPOSITION 2

**Proposition 5 (Proposition 2 restated)** *The supremum in in Equation 17 reaches its upper bound if and only if $\mathbb{P}_{\mathcal{Z}^{(1)},\ldots,\mathcal{Z}^{(M)},\mathcal{Y}} = \mathbb{G}$, where $\mathbb{G}$ is the Gibbs distribution defined as $d\mathbb{G} = \frac{e^T}{\mathbb{E}_{\mathbb{Q}}[e^T]} d\mathbb{Q}$ and $\mathbb{Q} = \mathbb{P}_{\mathcal{Z}^{(1)}} \times \cdots \times \mathbb{P}_{\mathcal{Z}^{(M)}} \times \mathbb{P}_{\mathcal{Y}}$.*

The supremum in Equation 17 reaches its upper bound when the gap is equal to 0,

$$\Delta \equiv D_{\mathrm{KL}}\left(\mathbb{P}\|\mathbb{Q}\right) - \mathbb{E}_{\mathbb{P}}[T] - \log\left(\mathbb{E}_{\mathbb{Q}}[e^T]\right) = 0, \tag{33}$$

where $\mathbb{P} = \mathbb{P}_{\mathcal{Z}^{(1)},\ldots,\mathcal{Z}^{(M)},\mathcal{Y}}$, $\mathbb{Q} = \mathbb{P}_{\mathcal{Z}^{(1)}} \times \cdots \times \mathbb{P}_{\mathcal{Z}^{(M)}} \times \mathbb{P}_{\mathcal{Y}}$. With the Gibbs distribution defined as $d\mathbb{G} = \frac{e^T}{Z} d\mathbb{Q}$, we have:

$$0 = D_{\mathrm{KL}}\left(\mathbb{P}\|\mathbb{Q}\right) - \mathbb{E}_{\mathbb{P}}\left[T\right] - \log\left(\mathbb{E}_{\mathbb{Q}}\left[e^T\right]\right) = D_{\mathrm{KL}}\left(\mathbb{P}\|\mathbb{G}\right), \tag{34}$$

where the second equality uses Equation 22. With Gibbs' inequality, we know $D_{\mathrm{KL}}\left(\mathbb{P}\|\mathbb{G}\right)$ equals 0 if and only if $\mathbb{P} = \mathbb{Q}$. Thus, we prove Proposition 5.

## A.4 PROOF OF PROPOSITION 3

**Proposition 6 (Proposition 3 restated)** *The two inequalities in Equation 27 simultaneously hold as equalities if and only if $\mathbb{P}_{\mathcal{X}^{(1)},\ldots,\mathcal{X}^{(M)},\mathcal{Y}} = \hat{\mathbb{G}}$, where $\hat{\mathbb{G}}$ is the Gibbs distribution defined as $d\hat{\mathbb{G}} = \frac{e^{F_\Theta}}{\mathbb{E}_{\mathbb{Q}}[e^{F_\Theta}]} d\mathbb{Q}$ and $\mathbb{Q} = \mathbb{P}_{\mathcal{X}^{(1)}} \times \cdots \times \mathbb{P}_{\mathcal{X}^{(M)}} \times \mathbb{P}_{\mathcal{Y}}$.*

Consider the TCMax loss:

$$\mathcal{L}_{\text{TCMax}} = -\mathbb{E}_{\mathbb{P}_{\mathcal{X}^{(1)},\dots,\mathcal{X}^{(M)},\mathcal{Y}}}\left[F_\Theta\right] + \log\left(\mathbb{E}_{\mathbb{P}_{\mathcal{X}^{(1)}}\times\cdots\times\mathbb{P}_{\mathcal{X}^{(M)}}\times\mathbb{P}_{\mathcal{Y}}}\left[e^{F_\Theta}\right]\right). \tag{35}$$

When two inequalities in Equation 27 simultaneously hold, we have:

$$
\begin{aligned}
\text{TC}(x^{(1)},\dots,x^{(M)},y) &= \sup_{T_{\mathcal{X}}:\Omega_{\mathcal{X}}\to\mathbb{R}}\mathbb{E}_{\mathbb{P}_{\mathcal{X}^{(1)},\dots,\mathcal{X}^{(M)},\mathcal{Y}}}\left[T_{\mathcal{X}}\right] - \log\left(\mathbb{E}_{\mathbb{P}_{\mathcal{X}^{(1)}}\times\cdots\times\mathbb{P}_{\mathcal{X}^{(M)}}\times\mathbb{P}_{\mathcal{Y}}}\left[e^{T_{\mathcal{X}}}\right]\right)\\
&= -\mathcal{L}_{\text{TCMax}} \\
&= \mathbb{E}_{\mathbb{P}_{\mathcal{X}^{(1)},\dots,\mathcal{X}^{(M)},\mathcal{Y}}}\left[F_\Theta\right] - \log\left(\mathbb{E}_{\mathbb{P}_{\mathcal{X}^{(1)}}\times\cdots\times\mathbb{P}_{\mathcal{X}^{(M)}}\times\mathbb{P}_{\mathcal{Y}}}\left[e^{F_\Theta}\right]\right).
\end{aligned} \tag{36}
$$

Hence, $F_\Theta$ reaches the upper bound of the supremum. With Proposition 5, we know Equation 36 holds if and only if $\mathbb{P}_{\mathcal{X}^{(1)},\dots,\mathcal{X}^{(M)},\mathcal{Y}} = \hat{\mathbb{G}}$. Thus, we prove the proposition.

## A.5 DERIVATION OF EQUATION 14

Since batch $\mathcal{B}$ is sampled according to the overall data distribution, we can consider the samples $(x_i^{(a)}, x_i^{(v)}, y_i)$, $\forall i \in \mathcal{B}$ as drawn from $\mathbb{P}_{\mathcal{A},\mathcal{V},\mathcal{Y}}$, where $x_i^{(a)}$, $x_i^{(v)}$, and $y_i$ follow the marginal distributions $\mathbb{P}_{\mathcal{A}}$, $\mathbb{P}_{\mathcal{V}}$, and $\mathbb{P}_{\mathcal{Y}}$, respectively. Since label distributions are generally assumed to be relatively uniform, we hypothesize $\mathbb{P}_{\mathcal{Y}}$ to be a uniform distribution over $\mathcal{Y}$. Thus, in calculations, $\mathbb{P}_{\mathcal{Y}}$ is directly treated as uniform without relying on batch sampling results. Substituting the assumptions into Equation 11 yields:

$$
\begin{aligned}
\mathcal{L}_{\text{TCMax}} = &-\frac{1}{|\mathcal{B}|}\sum_{i\in\mathcal{B}}\left(f_\theta\left(\psi_{\Theta_a}^{(a)}(x_i^{(a)}),\psi_{\Theta_v}^{(v)}(x_i^{(v)})\right)_{y_i}\right)\\
&+\log\left\{\frac{1}{|\mathcal{B}\times\mathcal{B}\times\mathcal{Y}|}\sum_{(j,k,y')\in\mathcal{B}\times\mathcal{B}\times\mathcal{Y}}\exp f_\theta\left(\psi_{\Theta_a}^{(a)}(x_j^{(a)}),\psi_{\Theta_v}^{(v)}(x_k^{(v)})\right)_{y'}\right\}\\
=&-\frac{1}{|\mathcal{B}|}\sum_{i\in\mathcal{B}}\log\frac{\exp f_\theta\left(\psi_{\Theta_a}^{(a)}(x_i^{(a)}),\psi_{\Theta_v}^{(v)}(x_i^{(v)})\right)_{y_i}}{\sum_{(j,k,y')\in\mathcal{B}\times\mathcal{B}\times\mathcal{Y}}\exp f_\theta\left(\psi_{\Theta_a}^{(a)}(x_j^{(a)}),\psi_{\Theta_v}^{(v)}(x_k^{(v)})\right)_{y'}} - \log|\mathcal{B}\times\mathcal{B}\times\mathcal{Y}|\\
=&-\frac{1}{|\mathcal{B}|}\sum_{i\in\mathcal{B}}\log\frac{\exp f_\theta\left(\psi_{\Theta_a}^{(a)}(x_i^{(a)}),\psi_{\Theta_v}^{(v)}(x_i^{(v)})\right)_{y_i}}{\sum_{(j,k,y')\in\mathcal{B}\times\mathcal{B}\times\mathcal{Y}}\exp f_\theta\left(\psi_{\Theta_a}^{(a)}(x_j^{(a)}),\psi_{\Theta_v}^{(v)}(x_k^{(v)})\right)_{y'}} - \log|\mathcal{B}|^2|\mathcal{Y}|,
\end{aligned} \tag{37}
$$

Thus, Equation 14 is obtained.

# B DETAILS OF EXPERIMENT

## B.1 DETAILS OF BASELINES

**Concatenation** Concatenation is a straightforward approach to multimodal fusion where the features from different modalities are combined by concatenating them into a single feature vector. In our experiments, we feed the combining vector into a single fully connected layer to get the prediction, which can be denoted as $f(X_i) = W[h_1(x_{i,1}),\cdots h_M(x_{i,M})]+b$. It can be decomposed as $f(X_i) = \sum_{m=1}^{M}\{W_m h_m(x_{i,m}) + b/M\}$ so can be simplified during training and define the output of modality $m$ is $f_m = W_m h_m(x_{i,m})+b/M$ following Peng et al. (2022).

**Share Prediction Head** This method uses a shared prediction head to calculate the output of every modality, then sums up all outputs from all modalities as the final output. Same as concatenation, we use a single fully connected layer as the shared head, and output can be denoted as $f(X_i) = \sum_{m=1}^{M}\{W h_m(x_{i,m}) + b/M\}$. Output of modality $m$ is defined as $f_m = W h_m(x_{i,m}) + b/M$.

**FiLM Perez et al. (2018)** FiLM modulates the feature from a modality using the feature from the other modality. Specifically, a FiLM layer performs a straightforward affine transformation on each feature of a neural network's intermediate representations, modulated by an arbitrary input. The output by FiLM is denoted as $f(X_i) = g\left(\gamma\left(h_2(x_{i,2})\right)\circ h_1(x_{i,1}) + \beta\left(h_2(x_{i,2})\right)\right)$, where $g$, $\gamma$ and $\beta$ are fully connected layers and $\circ$ here is the Hadamard product.

**Gated Kiela et al. (2018)**   Similar to FiLM, Gated uses the feature from one modality to modulate the other. The output of Gated is denoted as $f(X_i) = g\left(\gamma\left(h_1(x_{i,1})\right) \circ \sigma\left(\beta\left(h_2(x_{i,2})\right)\right)\right)$, where $g$, $\gamma$ and $\beta$ are fully connected layers and $\sigma$ is the sigmoid function. Notice the form of output by FiLM and Gated can not be decomposed into parts of modalities, so in our experiment, we deactivate a modality by inputting a zero tensor to get the single modality performance. In our experiment, audio modality is modality 1 and visual modality is modality 2 in the formula.

**Unimodal Learning**   Naive Unimodal Learning trains each modality separately and combines them during the prediction phase. Training separately helps the encoder of each modality efficiently learn modality-specific information. However, the model is unable to tell whether inputs from different modalities come from the same sample as modalities is inputted independently during training.

## B.2   Details of the Experiment on MVSA (Table 3)

In the experiment, to align with CLIP's prediction paradigm, we define class-specific features $c_y^{(i)}$ and $c_y^{(t)}$ for each class. For each label $y$, the model's output logit is computed as:

$$f_\theta(f_i, f_t) = \frac{s(c_y^{(i)}, f_i)}{\tau} + \frac{s(c_y^{(t)}, f_t)}{\tau} \tag{38}$$

where:

- $\theta = \{c_y^{(i)} | y \in \mathcal{Y}\} \cup \{c_y^{(t)} | y \in \mathcal{Y}\}$ is the trainable class-specific features
- $f_i$ and $f_t$ are CLIP's output features
- $\tau$ is CLIP's temperature coefficient
- $s(\cdot, \cdot)$ denotes cosine similarity

Training configuration:

- Total epochs: 100
- Batch size: 32
- Learning rate: 0.01 (decayed to 0.001 after epoch 70)

Due to the relatively small accuracy differences observed in the experiment, we include error bounds with 95% confidence intervals in Table 5. The results show that although the performance is very close, there is no overlap between the confidence intervals of multimodal accuracy (Multi) of TCMax and the second-best method (Joint Learning). This statistically significant difference ($p > 0.975^2 > 0.95$) confirms that the performance gap is not caused by experimental variance.

Table 5: Result of the average test accuracy(%) of 10 random seeds on MVSA with frozen CLIP pretrained encoders. We report the 95% confidence intervals.

| Methods | RN50 | | | ViT-B/32 | | |
|---|---|---|---|---|---|---|
| | Image | Text | **Multi** | Image | Text | **Multi** |
| Joint | $75.76 \pm 0.20$ | $73.60 \pm 0.16$ | $81.23 \pm 0.27$ | $76.88 \pm 0.09$ | $74.27 \pm 0.27$ | $82.83 \pm 0.15$ |
| Unimodal | $\mathbf{76.74 \pm 0.07}$ | $\mathbf{77.16 \pm 0.22}$ | $80.02 \pm 0.13$ | $\mathbf{78.54 \pm 0.09}$ | $\mathbf{76.97 \pm 0.07}$ | $81.77 \pm 0.15$ |
| TCMax | $75.38 \pm 0.12$ | $74.97 \pm 0.54$ | $\mathbf{81.75 \pm 0.23}$ | $78.03 \pm 0.16$ | $76.55 \pm 0.22$ | $\mathbf{84.05 \pm 0.15}$ |

## C   Potential in Regression Tasks

Although this paper primarily focuses on the task of multimodal image classification, TCMax may also be applied to other tasks, such as regression. Here, we use a simple regression task as an example to explore the potential of TCMax in regression scenarios.

We employ two sentiment analysis datasets: CMU-MOSI and CMU-MOSEI. We use MAG-BERT Rahman et al. (2020) as the baseline, which takes multimodal inputs from audio ($\mathcal{A}$), visual ($\mathcal{V}$), and text ($\mathcal{L}$) modalities and outputs a continuous value ($\mathcal{Y}$) representing the degree of positive sentiment. To train with TCMax we first define the mapping $F_\Theta : \mathcal{A} \times \mathcal{V} \times \mathcal{L} \times \mathcal{Y} \to \mathbb{R}$ in Equation 11.

We modified the prediction head to output both the predicted value and its confidence: $y_{pred} = y_{pred}(a, v, l)$ and $c_{pred}(a, v, l)$. We define $F_\Theta$ in our paper's Equation 11 as:

$$F_\Theta(a, v, l, y) = -\frac{(y_{pred}(a, v, l) - y)^2}{\sigma^2} + \lambda c_{pred}(a, v, l), \tag{39}$$

where $\sigma$ represents the standard deviation in the predicted Gaussian distribution, and $\lambda$ is a multiplicative coefficient to facilitate analysis. Let $\mathcal{B}$ and $\mathcal{B}_{\text{ns}}$ denote the batch and sampled negative set, respectively. Substituting this into Equation 11 yields:

$$
\begin{aligned}
\mathcal{L}_{\text{TCMax}} = &- \mathbb{E}_{(a,v,l,y) \in \mathcal{B}} \left[ -\frac{(y_{pred}(a, v, l) - y)^2}{\sigma^2} + \lambda c_{pred}(a, v, l) \right] \\
&+ \log \mathbb{E}_{(a,v,l,y) \in \mathcal{B}_{\text{ns}}} \left[ \exp \left( -\frac{(y_{pred}(a, v, l) - y)^2}{\sigma^2} + \lambda c_{pred}(a, v, l) \right) \right].
\end{aligned}
\tag{40}
$$

For classification tasks where $y$ takes discrete values, we conventionally assume a uniform distribution $\mathbb{P}_\mathcal{Y}$ since labels typically occur with roughly equal frequency. However, in this continuous $y$ case, we must explicitly define $y$'s probability distribution in $\mathcal{B}_{\text{ns}}$. For the derivation, we assume $y$ follows a uniform distribution over interval $[a, b]$, yielding the following loss function:

$$
\begin{aligned}
\mathcal{L}_{\text{TCMax}} = &\frac{1}{\sigma^2} \underbrace{\mathbb{E}_{(a,v,l,y) \in \mathcal{B}} \left[ (y_{pred}(a, v, l) - y)^2 \right]}_{\text{MSE}} \\
&+ \lambda \underbrace{\left\{ -\mathbb{E}_{(a,v,l,y) \in \mathcal{B}} \left[ c_{pred}(a, v, l) \right] + \log \mathbb{E}_{(a,v,l,y) \in \mathcal{B}_{\text{ns}}} \left[ \exp \left( c_{pred}(a, v, l) \right) \right] \right\}}_{\text{TCMax}} \\
&+ \log \int_a^b \exp \left( -\frac{(y_{pred}(a, v, l) - y)^2}{\sigma^2} \right) \mathrm{d}y - \log (b - a).
\end{aligned}
\tag{41}
$$

By taking the limits as $a \to -\infty$ and $b \to +\infty$ (i.e., extending to $\mathbb{R}$), while observing that the $-\log(b - a)$ term becomes divergent yet vanishes during differentiation (being a constant), we obtain the final form:

$$
\begin{aligned}
\mathcal{L}_{\text{TCMax}} = &\frac{1}{\sigma^2} \underbrace{\mathbb{E}_{(a,v,l,y) \in \mathcal{B}} \left[ (y_{pred}(a, v, l) - y)^2 \right]}_{\text{MSE}} \\
&+ \lambda \underbrace{\left\{ -\mathbb{E}_{(a,v,l,y) \in \mathcal{B}} \left[ c_{pred}(a, v, l) \right] + \log \mathbb{E}_{(a,v,l,y) \in \mathcal{B}_{\text{ns}}} \left[ \exp \left( c_{pred}(a, v, l) \right) \right] \right\}}_{\text{TCMax}} \\
&+ \lim_{a \to -\infty} \lim_{b \to +\infty} \log \int_a^b \exp \left( -\frac{(y_{pred}(a, v, l) - y)^2}{\sigma^2} \right) \mathrm{d}y \\
= &\frac{1}{\sigma^2} \underbrace{\mathbb{E}_{(a,v,l,y) \in \mathcal{B}} \left[ (y_{pred}(a, v, l) - y)^2 \right]}_{\text{MSE}} \\
&+ \lambda \underbrace{\left\{ -\mathbb{E}_{(a,v,l,y) \in \mathcal{B}} \left[ c_{pred}(a, v, l) \right] + \log \mathbb{E}_{(a,v,l,y) \in \mathcal{B}_{\text{ns}}} \left[ \exp \left( c_{pred}(a, v, l) \right) \right] \right\}}_{\text{TCMax}} + \log \sqrt{\pi} \sigma.
\end{aligned}
\tag{42}
$$

The resulting loss function naturally decomposes into two components: (1) The first term constrains regression accuracy (MSE); (2) The second term enforces TCMax's cross-modal alignment constraint on $(a, v, l)$ triplets. When $\lambda = 0$ (eliminating $c_{pred}$), the loss reduces to conventional MSE as the second term is eliminated.

Table 6 presents the experimental results on the CMU-MOSI and CMU-MOSEI datasets. After training with TCMax, the model achieves modest improvements on both datasets. This suggests that TCMax may hold potential research value for such regression tasks, and further in-depth studies will be conducted in the future.

Table 6: Result on CMU-MOSI and CMU-MOSEI datasets. For CMU-MOSI, we set $\sigma = 0.5$, while for CMU-MOSEI, $\sigma = 0.75$. In all experiments, $\lambda = 1$, and all results represent averages across three random seeds.

| Dataset | Mehod | Binary Acc↑ | F1↑ | MAE↓ | Corr↑ |
|---|---|---|---|---|---|
| CMU-MOSI | Baseline (MAG-BERT) | 83.36 | 83.21 | 0.7938 | 0.7644 |
| | TCMax | 84.27 | 84.13 | 0.7775 | 0.7753 |
| CMU-MOSEI | Baseline (MAG-BERT) | 85.14 | 85.10 | 0.5903 | 0.7867 |
| | TCMax | 85.61 | 85.52 | 0.5889 | 0.7887 |

