# OpenReview forum: "Multimodal Classification via Total Correlation Maximization"
_ICLR.cc/2026/Conference — ICLR 2026 Poster_

### Official Review · Reviewer_PGpr · 2025-10-24

**Soundness:** 4
**Presentation:** 3
**Contribution:** 3
**Rating:** 8
**Confidence:** 3

**Summary:**

- Addresses an important problem: competition between modalities in training by maximizing total correlation (across modalities + labels) instead of tweaking per-modality losses.
- Derives a lower bound (TCNE) and turns it into a practical, hyperparameter-free loss (TCMax)
- Drop-in training objective; no inference changes or extra heads/schedulers.
- Consistently boosts multimodal accuracy on audio-visual and image–text benchmarks; shows better cross-modal agreement (lower JS-divergence) and more balanced per-modality predictions.

**Strengths:**

- Clean, well motivated information-theoretic formulation, and nice conversion into a usable loss function with strong results
- Generally strong results across several datasets vs. recent balancing baselines.
- Compute-aware: sampling reduces forward passes
- Analysis beyond accuracy: JS-divergence and negative pair analysis are insightful

**Weaknesses:**

- Nit: Figure 2: an "illusion" -> illustration?
- See questions

**Questions:**

- How does this transfer across domains? because you're maximizing joint correlation with the labels on this dataset, it makes sense that you might fit the proper modality contributions here, but how do you know that you're not leading to modality competition in the transfer learning setting (which is common b/c this is probably going to be used for large scale pretraining)?
- What do you lose by sampling instead of doing all the forward passes theoretically required by the method?
- Do you have some explanation for why TCMax does not outperform baselines on unimodal? How would this fare in cases with high modality imbalance, where joint learning approaches unimodal learning?
- How would this the change in training loss function lead to changes in the realistic downstream uses of the model? e.g., using argmax predictions over logit distributions using distributions as a measure of confidence calibration?

---

> ### Author Response · Authors · 2025-11-23
>
> First, we would like to thank you for your recognition of our work. The typo in Figure 2 has been corrected in the updated version. Below, we address your questions one by one:
>
> ### 1. Q1: Transferability across Domains
>
> Modality competition typically occurs in joint learning scenarios because the two modalities compete for limited supervisory information. Our method has stronger supervision intensity than both joint learning and unimodal learning, thereby further reducing the risk of overfitting. Consequently, it exhibits relatively better generalization and theoretically superior transferability across domains. However, verifying this requires additional experiments, which we are unable to conduct due to time and computational resource constraints at present.
>
> ### 2. Q2: Cost of Sampling Negative Pairs
>
> Performing all forward passes for negative pairs is essentially to approximate the distribution $\mathbb{P} _{\mathcal{X}^{(1)}} \times \dots \times \mathbb{P} _{\mathcal{X}^{(M)}}$ in Equation 11. Since our data is also sampled from the overall data distribution, this approach only provides an approximation rather than an exact equivalence. The method of sampling negative pairs represents a compromise in the sampling process, merely switching from a higher-precision approximation to a lower-precision one, while the distributions being estimated remain consistent.
>
> As shown in Figure 3, fewer samples result in lower computational costs but at the expense of partial model accuracy.
>
> ### 3. Q3: TCMax not Outperforming Baselines on Unimodal Tasks
>
> This is because unimodal methods are equipped with sufficient supervision, enabling the models to be fully trained. In contrast, TCMax actually introduces joint learning-based supervision, which offers no benefit for evaluating unimodal scores and may even interfere with the training of unimodal supervisory information to a certain extent. This leads to TCMax's unimodal performance not being prominent compared to dedicated unimodal learning methods.
>
> TCMax is suitable for scenarios where the interaction between modalities can significantly enhance predictive capabilities. In cases of extreme modality imbalance, where one modality is too weak to assist the stronger one, TCMax's performance will be comparable to that of both joint learning and unimodal learning. Fortunately, in most cases, TCMax achieves performance no inferior to either joint learning or unimodal learning.
>
> ### 4. Q4: Changes in Downstream Uses
>
> This is an interesting question! There is a distinct difference in probabilistic modeling between the model trained with TCMax and those trained with joint learning or unimodal learning. As described in Proposition 3, the model trained with TCMax theoretically directly estimates $\mathbb{P} _{\mathcal{X}^{(1)} \times \cdots \times \mathcal{X}^{(M)} \times \mathcal{Y}}$. In contrast, joint learning models estimate $\mathbb{P} _{\mathcal{Y} | \mathcal{X}^{(1)} \times \cdots \times \mathcal{X}^{(M)}}$, while unimodal learning models estimate $\mathbb{P} _{\mathcal{Y} | \mathcal{X}^{(i)}}$.
>
> Since $\mathbb{P} _{\mathcal{Y} | \mathcal{X}^{(1)} \times \cdots \times \mathcal{X}^{(M)}}$ and $\mathbb{P} _{\mathcal{Y} \mid \mathcal{X}^{(i)}}$ can be directly derived from $\mathbb{P} _{\mathcal{X}^{(1)} \times \cdots \times \mathcal{X}^{(M)} \times \mathcal{Y}}$ and the data distribution, we believe that methods such as argmax predictions over logit distributions are also applicable to models trained with TCMax.

---

### Official Review · Reviewer_48tw · 2025-10-29

**Soundness:** 1
**Presentation:** 3
**Contribution:** 3
**Rating:** 2
**Confidence:** 5

**Summary:**

This paper proposes TCMax, a multimodal classification objective that consists a lower-bound of Total Correlation. To achieve this lower bound, the authors utilize the MINE of Belghazi and the Donsker-Varadhan representation. The idea of rebasing the expectation of $E_P$ with $Q$ being the factorized $P_{Z_1} P_{Z_2}..P_Y$ is quite interesting for the multimodal training.

**Strengths:**

The paper has a sound proof to derive the lower bound to TC and an interesting idea to compare the expectation of the joint multimodal distribution with the unimodal ones. The problem and the solution as well motivated.

**Weaknesses:**

There is one quite major issue. Following the code in the anonymized repo, it seems that the method is using the test set to select the best model during training. This falls under the data leakage between the validation/test set, which are the same in this case. With this major issue, we drive the paper for rejection. Despite that, I will provide some more input on the rest of the paper since the idea is well put.

The $I(Z_1;Y) + I(Z_2;Y) + I(Z_1, Z_2 | Y) = TC$ ignores that $Z_1, Z_2$ could correlate to predict $Y$, which essentially would be the synergy described by other papers. When you suggest that unimodal training and alignment beyond the task could achieve TC, you ignore this part. I consider this minor since you don't include it somehow in your method, if I understand it well. If you do, please elaborate.

The paper has not included a very important work that is quite close to the method proposed, MCR [1]. There are four parts that are worth discussing. First, they propose a different factorization of the joint mutual information (similarly to TC). Second, they showcase that supervised contrastive learning is lower-bound to some specific MI terms. I will come back to this. Third, they suggest that maximizing the CMI solely is not always the best option since you can still be stuck in a local minimum that you have optimized only one part of this. Finally, what you suggest is also a permutation of modalities and a penalization of their predictive probability which is quite close to permutation importance. Overall, I think comparing with this work and elaborating on improvements/differences will be highly necessary to illustrate the power and novelty of your paper.

About the supervised contrastive learning, what you suggest resembles an InfoNCE that has one positive and many negatives, even the ones with the same label. Including comparison with supervised contrastive or other ways to choose the positive/negative sets shall improve the understanding of the method. Additionally, commenting in terms of MI terms would be extra useful here.

An intuitive explanation of what the final loss tries to push for will help readers convey the final message.

Also, please include D\&R [2] for comparison since it has shown to be a strong baseline.

Lastly, I would like to incentivize scaling beyond the typical datasets and models. That should contribute further to our understanding of multimodal competition.

[1] Kontras, Konstantinos, et al. "Multimodal Fusion Balancing Through Game-Theoretic Regularization." arXiv preprint arXiv:2411.07335 (2024).

[2] Yake Wei, Siwei Li, Ruoxuan Feng, and Di Hu. Diagnosing and re-learning for balanced multimodal learning. In European Conference on Computer Vision, pages 71–86. Springer, 2024.

**Questions:**

Could you provide results with a separate validation/test set that doesnt include any form of data leakage, and explain this extensively somewhere in your supplementary matterial?

How does you method compare to MCR, D&R and supervised contrastive learning, both conceptually and experimentally?

Could you include an intuitive explanation of your method?

It has been shown that MINE suffers from high variance as an estimator of MI, do you face a similar issue?

---

> ### Author Response · Authors · 2025-11-23
>
> ### Q1: Data Leakage Problem
>
> Thank you sincerely for pointing out the experimental issue. We have revised the code and re-run the experiments in the updated manuscript (results for some datasets are pending due to time constraints, and updated results are marked in blue). However, we would like to provide additional context: this problem originated from the fact that our code was developed based on the OGM_GE codebase, and a considerable number of methods in this field (including D&R as you mentioned) exhibit the same issue.
>
> ### Weakness: Explanation of the decomposition $TC=I(Z_1;Y)+I(Z_2;Y)+I(Z_1,Z_2|Y)$
>
> > The $I(Z_1;Y)+I(Z_2;Y)+I(Z_1;Z_2|Y)=TC$ ignores that $Z_1, Z_2$ could correlate to predict $Y$, which essentially would be the synergy described by other papers. When you suggest that unimodal training and alignment beyond the task could achieve TC, you ignore this part. I consider this minor since you don't include it somehow in your method, if I understand it well. If you do, please elaborate.
>
> In our paper, the decomposition $TC=I(Z_1;Y)+I(Z_2;Y)+I(Z_1,Z_2|Y)$ is presented merely as a form of total correlation decomposition. In practice, we directly estimate the left-hand side (TC) during computation. Another decomposition of TC is $TC=I(Y;Z_1,Z_2)+I(Z_1;Z_2)$, which clearly indicates that TC inherently incorporates synergy. Specifically, $I(Z_1,Z_2|Y)=I(Z_1;Y|Z_2)+I(Z_2;Y)$ in the first decomposition implicitly contains the synergy term $I(Z_1;Y|Z_2)$, as $I(Z_2;Y)$ only captures unimodal information.
>
> ### Q2: Comparison to MCR and D&R
>
> The key difference between MCR and our work lies in their optimization objective: MCR maximizes $I(Z_1;Z_2;Y)$, while we maximize $TC(Z_1,Z_2,Y)$. Both the factorization of TC and modality permutation are employed to maximize TC or $I(Z_1;Z_2;Y)$ in experimental computations. We chose not to maximize $I(Z_1;Z_2;Y)$ primarily because $I(Z_1;Z_2;Y)$ can exceed $I(X_1;X_2;Y)$. For instance, when $I(X_1;X_2;Y)<0$, it is feasible to set model parameters such that $I(Z_1;Z_2;Y)=0>I(X_1;X_2;Y)$. Consequently, maximizing $I(Z_1;Z_2;Y)$ may cause the statistical relationships of the model's embeddings to deviate from those of the true distribution.
>
> In contrast, since $TC(Z_1,Z_2,Y)\leq TC(X_1,X_2,Y)$ holds universally, the optimal scenario for maximizing $TC(Z_1;Z_2;Y)$ is $TC(Z_1;Z_2;Y)=TC(X_1,X_2,Y)$. Furthermore, each term in the two decompositions in Eq. 6 satisfies the following inequalities (which can be proven by analogy to Proposition 1):
>
> - $I(y;z^{(a)},z^{(v)})\leq I(y;x^{(a)},x^{(v)})$
> - $I(z^{(a)}; z^{(v)})\leq I(x^{(a)}; x^{(v)})$
> - $I(y; z^{(a)})\leq I(y; x^{(a)})$
> - $I(y; z^{(v)})\leq I(y; x^{(v)})$
> - $I(z^{(a)},z^{(v)}|y)\leq I(x^{(a)},x^{(v)}|y)$
>
> Thus, when $TC(z^{(a)},z^{(v)},y)=TC(x^{(a)},x^{(v)},y)$, all the above inequalities become equalities. Maximizing TC therefore simultaneously optimizes all the objectives corresponding to the decompositions mentioned above—meaning TC captures joint dependencies, alignment, and unimodal label dependencies, as stated in our paper. However, maximizing ternary mutual information does not possess this property.
>
> Incidentally, Eq. 5 in MCR is presented as:
> $$
> I(X_1;X_2;Y)=I(X_1;Y|X_2)+I(X_2;Y|X_1)+I(X_1;X_2)-I(X_1;X_2|Y)
> $$
> This decomposition is incorrect, likely a typo. In reality, they are maximizing $I(Z_1,Z_2;Y)$, as the correct form of the decomposition is:
> $$
> I(X_1,X_2;Y)=I(X_1;Y|X_2)+I(X_2;Y|X_1)+I(X_1;X_2)-I(X_1;X_2|Y)
> $$
> This is essentially the same as joint learning, which also maximizes $I(Z_1,Z_2;Y)$, leading to some confusion on our part.
>
> Regarding D&R: D&R improves the model training process, while TCMax focuses on the training objective—these two approaches are complementary and can coexist. In fact, they can be used simultaneously, though the effectiveness requires experimental verification.
>
> You mentioned supervised contrastive learning: InfoNCE is a lower bound estimator for mutual information (binary TC), while TCMax is a lower bound estimator for TC. Both are contrastive learning methods. The key difference is that TCMax uses n-tuples (e.g., the triplet of visual, audio, and label) as sample pairs, rather than the binary pairs in InfoNCE. In summary, TCMax is essentially a supervised contrastive learning method, more closely related to InfoNCE as an extension, rather than an unrelated contrastive approach.
>
> The performance of MCR and D&R on the CREMA-D and AVE datasets is shown below. Due to time constraints, we have not tested other datasets yet and will update the results in future revisions.
>
> | Method         | CREMA-D | AVE  |
> | -------------- | ------- | ---- |
> | Joint Learning | 66.2    | 60.4 |
> | D&R            | 71.5    | 61.1 |
> | MCR            | 75.6    | 63.5 |
> | TCMax(Concat)  | 77.6    | 63.2 |

---

> ### Author Response · Authors · 2025-11-23
>
> ### Q3: Intuitive Explanation of the Method
>
> Our method is inherently straightforward: we construct the TCMax loss, and training with this loss maximizes $TC(z^{(a)},z^{(v)},y)$. Ideally, after training, the model will satisfy $TC(z^{(a)},z^{(v)},y)\approx TC(x^{(a)},x^{(v)},y)$, which makes the five inequalities mentioned in Q2 approximately hold as equalities. This imposes strict constraints on the model, reducing the risk of overfitting.
>
> We have made every effort to present our method intuitively in the paper, but we acknowledge there is room for improvement. We would greatly appreciate it if you could specify which parts lack clarity, and we will revise them accordingly.
>
> ### Q4: High Variance as an Estimator
>
> This is an interesting observation, and we agree that high variance may indeed be a concern—TCNE is essentially a multivariate extension of MINE, after all. However, in our paper, TCNE is used to optimize the model rather than to accurately estimate TC. Similar to how InfoNCE serves as a lower bound estimator for MI, researchers typically focus on its performance in training models rather than its estimation accuracy.

---

### Official Review · Reviewer_K9M3 · 2025-10-31

**Soundness:** 3
**Presentation:** 4
**Contribution:** 3
**Rating:** 6
**Confidence:** 4

**Summary:**

this paper tackles multimodal supervised learning using an information theory perspective. It shows why the modality competition occurs classically when optimizing the cross-entropy loss in a joint-learning framework and advocates for a different strategy: total correlation maximization between modalities and the target. The model is evaluated on six multimodal classification benchmarks and two regression benchmarks, and it demonstrates state-of-the-art results across all datasets.

**Strengths:**

-	The paper is well written and clearly explains the motivation for their method from an information theory perspective.
-	The mathematical analysis is easy to follow, and the proposed loss is simple, yet effective.
-	The evaluation of the proposed method against baselines is fair and thorough, spanning across different domains and modalities (images, audios, videos).
-	The results are strong in most cases and clearly confirms the hypothesis made by the authors regarding the modeling of cross-modal interactions.

**Weaknesses:**

-	While I appreciate the completeness in the evaluation protocol in Table 2, I think it neglects the recent emergence of foundation models in the field of unimodal or multimodal representation learning (such as CLIP for vision and language, DINOv3 for vision, Wav2vec 2.0 for speech). I think it is also important to consider these pre-trained models as feature extractors and to apply your method on top of these, as you did afterwards in Table 4 for another dataset. It would clearly demonstrate the benefit of TCMax in a real-life scenario.
-	Supervised multimodal learning is a bit restrictive in terms of applications. Large-scale multimodal data come usually with very few annotations. I wonder how the proposed model would perform in the case of few-shot learning of semi-supervised learning.
-	Baseline models and concurrent approaches: since this work is deeply rooted in information theory, I think the works by Paul Pu Liang need to be properly cited and added to the baselines, for instance [1]. His work on self-supervised and supervised multimodal learning (using for instance the Partial Information Decomposition approach [2]) is very close to the one developed in this paper.
-	You mentioned: « without loss of generality, we analyze the scenario with two modalities (audio and visual) »: from an information theory perspective, it restricts the analysis very much. As the authors mentioned, the MI between n=3 variables can be negatives (not the case when n=2), quantifying the interactions generally between n>2 variables is hard (see for instance the Partial Information Decomposition theory, O-Information, gradient of O-Information, etc…) and Total Correlation gives you only a very broad measure of independence between your input variables (without telling anything about  the interactions between paired or triplet of variables in a general system of n variables). I would expect at least a reformulation of this sentence, if not a discussion about it at the end (along with the other limitations of your work).
-	Typos: Figure 2 “Illusions” -> ”Illustration”

[1] Learning factorized multimodal representations, Tsai et al., ICLR 2019
[2] Quantifying & Modeling Multimodal Interactions: An Information Decomposition Framework, Liang et al., NeurIPS 2023

**Questions:**

-	In equation 6, can you clarify why optimizing I(za, zv |y) is useful in your case since it quantifies the information contained in za and zv irrelevant for y. I would expect this term to actually decrease during training.
-	Implementation details: what is the architecture of the prediction (fusion) head? How did you choose it? Is it similar to other baselines? Does the architecture impact the results?
-	Are you going to release your code for reproducibility?
-	You mentioned the computation cost of your method (at least during training). Did you quantify it in practice ?

---

> ### Author Response · Authors · 2025-11-23
>
> ### W1: Experiments on More Pretrained Models
>
> Thank you for your valuable suggestion. The current experiments only use relatively small models and datasets due to two main considerations:
>
> 1. Unfortunately, training larger models is challenging for us due to computational resource constraints.
> 2. After adopting pretrained models, the strong inherent capabilities of these models significantly alleviate modality competition on the small datasets used in the current experiments. This makes it difficult to compare the performance of different methods (as their results are relatively close, e.g., the performance of various methods in Table 4 is very similar). In such cases, larger-scale datasets are required.
>
> We sincerely appreciate your recommendation. In future work, we will explore our method on more practical tasks (such as the regression tasks mentioned in Appendix C). Since these tasks are more difficult for models to learn, we will employ larger models at that time.
>
> ### W2: Performance on Few-Shot Learning or Semi-Supervised Learning
>
> The current form of TCMax is not suitable for few-shot learning or semi-supervised learning. As a purely supervised learning loss function, its learning objective heavily relies on $\mathbb{P_{\mathcal{X}^{(a)} \times \mathcal{X}^{(v)} \times \mathcal{Y}}}$, which includes the distribution of labeled data $\mathcal{Y}$. Adapting TCMax to these two tasks would require additional design, and we do not anticipate that TCMax will outperform other methods in these scenarios at present.
>
> ### W3: More Baselines
>
> Thank you for proposing these two baselines. However, we do not plan to supplement these experiments for the following reasons:  First, these methods focus more on representation learning, and their training objectives are not solely optimized for classification performance. Thus, it is difficult to compare their actual performance with other methods in the paper; Second, due to personal time constraints, we are unable to conduct too many additional experiments. Thank you for your understanding.
>
> Nevertheless, another reviewer mentioned FactorCL-SUP, which belongs to the same category as the two baselines you proposed. We have conducted experiments on FactorCL-SUP using the CREMA-D and AVE datasets, but its performance is not ideal:
>
> | Method         | CREMA-D | AVE  |
> | -------------- | ------- | ---- |
> | Joint Learning | 66.2    | 60.4 |
> | FactorCL-SUP   | 51.7    | 41.8 |
> | TCMax(Concat)  | 77.6    | 63.2 |
>
> ### W4: Additional Analysis for $M>2$
>
> Thank you for your comment. The relevant formulation has been revised accordingly. We supplement the analysis for $M>2$ as follows:
>
> The analysis of joint learning and unimodal learning can be easily extended to scenarios with $M>2$ modalities. For joint learning, the decomposition for multiple modalities is:
> $$
> I(y;Z) = I(y; z^{(1)}, z^{(2)},\cdots,z^{(M)}) = I(y; z^{(i)}) + I(y; z^{(1)},\cdots,z^{(i-1)},z^{(i+1)},\cdots,z^{(M)} | z^{(i)}).
> $$
> Similarly, when $I(y; z^{(i)})$ is close to $H(y)$, the inequality $I(y;z^{(j)}|z^{(i)})\leq I(y; z^{(1)},\cdots,z^{(i-1)},z^{(i+1)},\cdots,z^{(M)} | z^{(i)})$ holds for all $j\neq i$. As analyzed in our paper, the right-hand side of this inequality is small, so the conclusion regarding the occurrence of modality competition remains valid.
>
> For unimodal learning, extending to any number of modalities does not affect the learning of individual modalities, and there is no interaction between modalities. Thus, the conclusion is consistent.
>
> For TCMax, the formulation is analogous but requires replacing the conditional mutual information with the higher-dimensional conditional total correlation:
> $$
> TC(z^{(1)}, \cdots, z^{(M)}, y) = I(y; z^{(1)},\cdots, z^{(M)}) +  \text{TC}(z^{(1)}; \cdots ; z^{(M)})
> $$
> $$
> TC(z^{(1)}, \cdots, z^{(M)}, y) =I(y; z^{(1)}) + \cdots + I(y; z^{(M)}) + \text{TC}(z^{(1)}, \cdots, z^{(M)} | y)
> $$
> Here, the conditional total correlation is defined as:
>
> $$
> TC(z^{(1)}, \cdots, z^{(M)} | y) = D _{KL} \left( \mathbb{P} _{z^{(1)}, \dots, z^{(M)} | y}  \prod _{k=1}^M  \mathbb{P} _{ z^{(k)} | y}   \right)
> $$

---

> ### Author Response · Authors · 2025-11-23
>
> ### Q1: The Role of $I(z^{(a)};z^{(v)}|y)$
>
> Your intuition is correct. When the model can be well-trained solely under the supervision of $I(z^{(a)},z^{(v)};y)$, $I(z^{(a)};z^{(v)}|y)$ may instead interfere with the main training objective $I(z^{(a)},z^{(v)};y)$. However, the scenario we currently discuss involves modality competition, which indicates that the model is prone to overfitting on the training set and thus insufficiently trained under the supervision of $I(z^{(a)},z^{(v)};y)$. In this case, introducing $I(z^{(a)};z^{(v)}|y)$ as an additional training objective can prevent the model from overfitting prematurely. Furthermore, increasing $I(z^{(a)};z^{(v)}|y)$ enables the strong modality to assist the learning process of the weak modality encoder.
>
> ### Q2: The Architecture of the Fusion Head
>
> We validated TCMax using two types of fusion heads: share head and concat. This choice is simply because these two fusion heads are the simplest and are used in the comparative methods in our paper. Specifically, the comparative methods using the concat fusion head include OGM-GE, OPM, AGM, QMF, and MMPareto, while MLA uses the share head.
>
> The fusion head does affect the final results. In fact, the more complex the fusion head, the more likely the model is to overfit, thereby reducing its generalization ability.
>
> ### Q3: Release Code for Reproducibility
>
> Yes! We plan to make the code publicly available after organizing it. An anonymous repository link is provided in the abstract of the paper, where you can access our initial code.
>
> ### Q4: Computation Cost of TCMax
>
> TCMax does incur additional computational overhead, which can be directly calculated. According to the computation method in Eq. 15, the main additional overhead comes from $|\mathcal{N}|$ forward passes of the fusion head (the overhead of computing the loss itself is negligible and thus omitted here). For the concat and share head fusion heads used in this paper, further optimization can be achieved using the method in Eq. 16, which avoids these $|\mathcal{N}|$ forward passes and thus introduces almost no additional overhead.

---

### Official Review · Reviewer_TfiU · 2025-10-31

**Soundness:** 3
**Presentation:** 3
**Contribution:** 3
**Rating:** 6
**Confidence:** 3

**Summary:**

This paper addresses the problem of modality competition in multimodal learning, where multimodal models tend to overfit dominant modalities and underutilize weaker ones, sometimes performing worse than unimodal baselines. While previous methods have attempted to rebalance or combine joint and unimodal learning empirically, this work takes an information-theoretic approach. The authors analyze modality competition through the lens of total correlation.

The authors propose a new method for multimodal classification that maximizes total correlation between multimodal features and target labels, which naturally promotes more balanced learning and integrates multimodal interactions. They build on Mutual Information Neural Estimation (MINE) and introduce Total Correlation Neural Estimation (TCNE), which estimates a lower bound on total correlation. Using this, they develop TCMax, a novel loss function that optimizes total correlation via a variational bound. Experimental results reportedly show that TCMax outperforms both joint and unimodal baselines, on several datasets.

While the method is currently limited to fully supervised classification tasks, the proposed theoretical framework is both rigorous and well-justified, offering valuable insights that could inspire broader multimodal learning research.

**Strengths:**

1. Strong theoretical foundation: The paper provides a clear and rigorous information-theoretic formulation for learning from multimodal inputs.

2. Novel objective function: The introduction of Total Correlation Neural Estimation (TCNE) and the TCMax loss.

3. Conceptual clarity: The theoretical motivation is well-grounded and is easily applied into practice.

4. Empirical validation: The experimental results consistently demonstrate performance improvements over multimodal and unimodal baselines.

**Weaknesses:**

1. **Vague definition of weak and strong modalities.**
Although the paper discusses modality competition, the criteria used to define or quantify “weak” versus “strong” modalities are not clearly specified. Providing a more explicit operational definition or empirical measure would strengthen the theoretical analysis and clarify the interpretation of the results. Additional experiments that explicitly quantify these distinctions would further support the claims.

2. **The claim that the analysis with two modalities holds “without loss of generality” is not justified (l. 141).** In multimodal settings with $M>2$ higher-order dependencies (synergy, redundancy) emerge that are irreducible to pairwise terms. Consequently, the total-correlation objective, identifiability conditions, and estimation behavior differ qualitatively from the bimodal case. Any guarantees or intuitions derived for two modalities therefore cannot be presumed to extend to
$M>2$ without additional analysis.

3. **In the same vibe as the previous point, scalability to multiple modalities is not clear.**
The current formulation and experiments seem primarily focused on bimodal settings. It is not obvious how the proposed total correlation maximization framework extends to scenarios involving more than two modalities, where inter-modal dependencies become more complex.

4. **Restriction to supervised learning.**
The approach assumes access to fully labeled data, limiting its applicability to self-supervised multimodal scenarios, settings that are highly relevant in practice and that have been addressed in previous works [1, 2].


5. **Missing discussion of relevant related work (FactorCL, CoMM).**
The paper could more clearly articulate how its information-theoretic perspective relates to or differs from recent approaches that explicitly model shared and modality-specific information, such as FactorCL [1] (which uses mutual information–based decomposition) and CoMM [2] (which leverages partial information decomposition). A comparative discussion, both conceptual and empirical, would clarify the novelty and positioning of the proposed framework within this emerging research direction.

[1] Liang, P. P., Deng, Z., Ma, M. Q., Zou, J. Y., Morency, L. P., & Salakhutdinov, R. (2023). Factorized contrastive learning: Going beyond multi-view redundancy. Advances in Neural Information Processing Systems, 36, 32971-32998.

[2] Dufumier, B., Castillo-Navarro, J., Tuia, D., & Thiran, J. P. (2025). What to align in multimodal contrastive learning?. International Conference on Learning Representations.

**Questions:**

- How do the authors quantitatively define “weak” and “strong” modalities in their analysis?

- The paper claims that analyzing two modalities holds “without loss of generality.” Could the authors clarify the theoretical justification for this claim? How does the proposed framework account for higher-order dependencies (e.g., synergy, redundancy) that arise when $M > 2$ Would the total correlation objective or optimization strategy require modifications?

- Is the proposed total correlation objective compatible with or could be adapted for self-supervised learning setups?

---

> ### Author Response · Authors · 2025-11-23
>
> ### W1 & Q1: Definition of Weak and Strong Modalities
>
> Thank you for pointing out the issue with our formulation. The term "Weak and Strong Modalities" in our paper is used to distinguish the relative learning effectiveness between imbalanced modalities in the context of modality imbalance. Following the definition in [1], a modality with higher unimodal classification accuracy is regarded as a strong modality, while the one with relatively lower unimodal accuracy is defined as a weak modality.
>
> As shown in Table 1 of our paper, the Concat and Share Head methods (both belonging to joint learning) typically yield significant discrepancies in prediction accuracy between the two modalities. In such cases, the modality with higher accuracy is referred to as the strong modality, and the one with lower accuracy as the weak modality.
>
> ### W2 & W3 & Q2: Additional Analysis for $M>2$
>
> Thank you for your valuable comment. The relevant formulation has been revised accordingly.
>
> The analysis of joint learning and unimodal learning can be easily extended to scenarios with $M>2$ modalities. For joint learning, the decomposition for multiple modalities is expressed as:
>
> $$
> I(y;Z) = I(y; z^{(1)}, z^{(2)},\cdots,z^{(M)}) = I(y; z^{(i)}) + I(y; z^{(1)},\cdots,z^{(i-1)},z^{(i+1)},\cdots,z^{(M)} | z^{(i)}).
> $$
>
> Similarly, when $I(y; z^{(i)})$ is close to $H(y)$, the inequality $I(y;z^{(j)}|z^{(i)})\leq I(y; z^{(1)},\cdots,z^{(i-1)},z^{(i+1)},\cdots,z^{(M)} | z^{(i)})$ holds for all $j\neq i$. As analyzed in our paper, the right-hand side of this inequality is small, so the conclusions in the manuscript remain valid.
>
> For unimodal learning, extending to any number of modalities does not affect the learning of individual modalities, and there is no interaction between modalities. Thus, the conclusions are also consistent.
>
> For TCMax, the formulation is analogous but requires replacing the conditional mutual information with the higher-dimensional conditional total correlation:
>
> $$
> TC(z^{(1)}, \cdots, z^{(M)}, y) = I(y; z^{(1)},\cdots, z^{(M)}) +  \text{TC}(z^{(1)}; \cdots ; z^{(M)})
> $$
> $$
> TC(z^{(1)}, \cdots, z^{(M)}, y) =I(y; z^{(1)}) + \cdots + I(y; z^{(M)}) + \text{TC}(z^{(1)}, \cdots, z^{(M)} | y)
> $$
>
> Here, the conditional total correlation is defined as:
>
> $$
> TC(z^{(1)}, \cdots, z^{(M)} | y) = D _{KL} \left( \mathbb{P} _{z^{(1)}, \dots, z^{(M)} | y}  \prod _{k=1}^M  \mathbb{P} _{ z^{(k)} | y}   \right)
> $$
>
> Regarding experiments with $M>2$ modalities, due to the scarcity of three-modal datasets, we have limited experimental results in this area. However, in Appendix C, we provide experimental results on the CMU-MOSI and CMU-MOSEI datasets for regression tasks. Although comparisons with other methods are lacking, our approach achieves a certain degree of improvement over the baseline. Currently, our method has been thoroughly validated primarily on bimodal datasets.

---

> ### Author Response · Authors · 2025-11-23
>
> ### W4 & Q3: Performance on Self-Supervised Multimodal Scenarios
>
> At present, our research is limited to the classification task. We will extend our work to other tasks (such as regression or self-supervised learning) in future studies.
>
> TCMax has the potential to be applied to self-supervised learning scenarios. We treat the label as an additional modality in the paper, followed by contrastive learning among three modalities (audio, visual, and label). Thus, if the label is also considered as an input, TCMax can be regarded as a self-supervised loss.
>
> However, self-supervised learning experiments require large amounts of data and computational resources. Due to computational constraints, we are unable to complete these experiments in the short term. Thank you for your suggestion; we will consider the feasibility of applying TCMax to self-supervised training in our future research.
>
> ### W5: Related Work
>
> Thank you for mentioning the FactorCL and CoMM methods. However, we believe that except for FactorCL-SUP, the other methods should not be included as comparative baselines in this paper.
>
> FactorCL-SSL and CoMM primarily aim to address the issue of only learning redundant information in contrastive learning, so they are essentially self-supervised contrastive learning methods. During training, they focus on representation learning and then adapt to downstream tasks by fitting a classification head. Therefore, without a pre-trained dataset, these methods are difficult to compare with methods that directly focus on classification performance (such as TCMax and other methods in our experiments).
>
> The performance of FactorCL-SUP on the CREMA-D and AVE datasets is shown below:
>
> | Method         | CREMA-D | AVE  |
> | -------------- | ------- | ---- |
> | Joint Learning | 66.2    | 60.4 |
> | FactorCL-SUP   | 51.7    | 41.8 |
> | TCMax(Concat)  | 77.6    | 63.2 |
>
> It can be seen that FactorCL-SUP does not perform well. Its optimization objective is to maximize $I(z^{(a)},z^{(v)}|y)$, which is consistent with joint learning. However, the introduction of multiple decomposition forms—each involving estimators—leads to complex interactions between these estimators. As a result, FactorCL-SUP is less effective than joint learning (a more direct method for maximizing $I(z^{(a)},z^{(v)}|y)$) in achieving this single objective.

---

### Meta-Review · Area_Chair_oQoH · 2025-12-19

**Summary:**

This paper analyzes the modality competition problem in multi-modal learning from information-theoretic perspective and proposes to maximize total correlation between multi-modal features and ground truth labels instead of employing uni- or multi-modal training solely or simultaneously. Based on this concept, it introduces a TCMax loss for supervised multi-modal learning which aims to maximize the Total Correlation Neural Estimation. Extensive experiments showcase the effectiveness of proposed TCMax loss and validate the theoretical analysis.

**Reviewer Concerns:**

Concerns of reviewers include the following: 1) scalability of TCMax loss beyond 2 modalities (reviewer TfiU, K9M3, and 48tw); 2) restricted in the supervised learning paradigm (reviewer TfiU, K9M3); 3) missing baselines and related works (reviewer TfiU, K9M3, and 48tw); 4) data leakage in experiments leading to unreliable results (reviewer 48tw); 5) impact of pre-trained encoder (reviewer PGpr); 6) cross-domain generalization of the proposed method, and whether it has an impact on downstream applications (reviewer PGpr).

The authors have addressed most of these concerns during rebuttal. Specifically, the authors made the following efforts: 1) theoretically validated the feasibility of the proposed method in multi-modal (>2 modalities) scenarios and conducted additional experiments on the regression task (even though the improvement is marginal compared with the baseline); 2) made enough discussion of related works and included necessary baseline to compare; re-ran most experiments and pointed out the root of wrong implementation; 3) added experiment to explore different pre-trained encoder (e.g., CLIP visual encoders); 4) convey discussion on the transferibility of TCMax loss and its impact on downstream applications. Nevertheless, the authors acknowledged the limited scope of TCMax loss within supervised learning, which means the mitigating modality completion in semi- or self-supervised multi-modal representation learning is left for future work.

**Reviewer Scores:**

If the reviewers had been able to participate fully in the discussion, I think reviewer 48tw might have raised the score, since the major concern could be addressed by re-running most experiments. As for others, I think they would keep the original score.

---

### Decision · Program_Chairs · 2026-01-26

Accept (Poster)